# Visual perception of texture regularity: Conjoint measurements and a wavelet response-distribution model

**Hua-Chun Sun**[1,2], **David St-Amand**[1], **Curtis L. Baker, Jr.**[1]*, **Frederick A. A. Kingdom**[1]

**1** McGill Vision Research, Department of Ophthalmology, McGill University, Montreal, Canada, **2** School of Psychology, UNSW Sydney, Australia

* curtis.baker@mcgill.ca

**Data Availability Statement:** All relevant data are within the manuscript and its Supporting Information files.

**Funding:** This project was supported by Natural Sciences and Engineering Research Council of

## Abstract

Texture regularity, such as the repeating pattern in a carpet, brickwork or tree bark, is a ubiquitous feature of the visual world. The perception of regularity has generally been studied using multi-element textures in which the degree of regularity has been manipulated by adding random jitter to the elements' positions. Here we used three-factor Maximum Likelihood Conjoint Measurement (MLCM) for the first time to investigate the encoding of regularity information under more complex conditions in which element spacing and size, in addition to positional jitter, were manipulated. Human observers were presented with large numbers of pairs of multi-element stimuli with varying levels of the three factors, and indicated on each trial which stimulus appeared more regular. All three factors contributed to regularity perception. Jitter, as expected, strongly affected regularity perception. This effect of jitter on regularity perception is strongest at small element spacing and large texture element size, suggesting that the visual system utilizes the edge-to-edge distance between elements as the basis for regularity judgments. We then examined how the responses of a bank of Gabor wavelet spatial filters might account for our results. Our analysis indicates that the peakedness of the spatial frequency (SF) distribution, a previously favored proposal, is insufficient for regularity encoding since it varied more with element spacing and size than with jitter. Instead, our results support the idea that the visual system may extract texture regularity information from the moments of the SF-distribution across orientation. In our best-performing model, the variance of SF-distribution skew across orientations can explain 70% of the variance of estimated texture regularity from our data, suggesting that it could provide a candidate read-out for perceived regularity.

## Author summary

We investigated human perception of texture regularity, in which subjects made comparative judgements of regularity in pairs of texture stimuli with differing levels of three parameters of texture construction—spacing and size of texture elements, and their positional jitter. We analyzed the data using a novel approach involving three-factor

Canada (https://www.nserc-crsng.gc.ca/index_eng.
asp) grants to CB (RGPIN-2017-05292) and to FK
(RGPIN-2016-03915). HCS was supported by a
fellowship funded by the Research Institute of the
McGill University Health Centre (https://rimuhc.ca/)
and a scholarship funded by the McGill University
Health Centre Foundation (https://muhcfoundation.
com/). The funders had no role in study design,
data collection and analysis, decision to publish, or
preparation of the manuscript.

**Competing interests:** The authors have declared
that no competing interests exist.

Maximum Likelihood Conjoint Measurement (MLCM). We utilized a novel three-way approach in MLCM to evaluate the effect size and significance of the three factors as well as their interactions. We found that all three factors contributed to perceived regularity, with significant main effects and interactions between factors, in a manner suggesting edge-to-edge distances between elements might contribute importantly to regularity judgments. Using a bank of Gabor wavelet spatial filters to model the response of the human visual system to our textures, we compared four types of ways that the distribution of wavelet responses could account for our measured data on perceived regularity. Our results suggest that the orientation as well as spatial frequency (SF) information from the wavelet filters also contributes importantly—in particular, the skew of the variance of the SF-distribution across orientation provides a candidate basis for perceived texture regularity.

## Introduction

Naturally occurring textures in the world vary in a seeming multitude of feature dimensions, and how they are represented in the human visual system remains poorly understood. Texture regularity (henceforth, "regularity"), defined as the degree of orderliness of element positions in a texture, is a ubiquitous feature of the visual world, seen in the repetitive patterns of carpets, brickwork, or tree bark, for example, and in experimental stimuli consisting of uniform micro-patterns in a quasi-periodic arrangement [1,2]. It thus carries crucial information about the structure, etiology and biological function of both natural and artificial surfaces [3,4]. Texture regularity is an elemental perceptual property that is adaptable [5,6] and can be affected through lateral inhibition by a surrounding texture [7]. Regularity also has been found to influence several types of visual processing such as texture segregation [8], contour detection [9], numerosity perception [10–12] and surface slant perception [1]. However, little is known about how the visual system encodes and represents regularity information, given its importance in visual perception.

Most experimental studies of regularity have applied random positional perturbation ("jitter") independently to the horizontal and vertical element positions in a notional lattice pattern of texture elements to construct textures of varying regularity levels–a larger range of the jitter in each element position gives more irregular textures [5–7,13]. It is believed that the representation of pattern regularity can be encoded rapidly and in a parallel manner at an early stage of visual processing, without requiring the encoding of individual element position [14,15]. This idea is supported by evidence from the "dipper" effect of discrimination thresholds found in imperfectly regular patterns [14], as well as the regularity aftereffect, in which perceived regularity can be altered through adaptation [5,6], in a manner which is insensitive to retinal location [5].

In line with the above studies, researchers have suggested that the peakedness of the distribution of oriented spatial frequency-tuned filter responses might help explain regularity encoding [5,6,16]. Based on these studies as well as evidence from our simultaneous regularity contrast study [7], we suggested that in addition to SF-peakedness, other statistics of the SF-distribution–such as the bandwidth (width at half maximum, FWHM), kurtosis, skew and standard deviation–are also good predictors of perceived regularity, and could therefore be used for coding regularity in the visual system. Moreover, the variance of the SF-distribution across orientation could also be a potential read-out for regularity perception [7], since the regularity aftereffect is orientation selective [5].

Based on these SF models, it is expected that other factors which change the SF-distribution parameters, in addition to the widely used jittering of element position [5–7,13,14], may

potentially affect regularity perception. For example, increasing element size can increase SF-peakedness substantially, and as expected, regularity discriminability is enhanced [16]. Element spacing can also affect SF-peakedness, but the peak in the SF-response distribution related to spacing might overlap with the peak from size, giving two separate peaks [16] or a composite peak if they overlap. Consequently an account of regularity perception in terms of simple SF-peakedness as read-out may be insufficient, and other response distribution features may also contribute [7].

Previous studies of regularity perception mostly utilized jitter of element positions as the main manipulation [5–7,13,14] and the results corresponded well with simple SF-peakedness [5,6,16]. In this study we investigated the representation and encoding of regularity information considering other factors in addition to element jitter, and utilize a different approach–the Maximum Likelihood Conjoint Measurement (MLCM) method [17,18]. Conjoint measurement has been widely used to quantify how two physical stimulus dimensions contribute to a perceptual scale [17–23]. Gerardin et al. [24] considered three factors in MLCM, but the experimental design and statistical analysis are essentially 2-dimensional MLCM, performed separately for each of the three pairwise combinations of the three factors. This approach however does not reveal possible interactions between the three factors. Here we used conjoint measurement to investigate simultaneously the contributions and interactions between *three* features, element spacing, size and jitter, on perceived regularity, to better understand how the visual system jointly processes different sources of information to achieve texture regularity perception. We used the psychophysical method of paired comparisons to provide the data for the MLCM analysis. On each trial two texture stimuli were presented with different combinations of element spacing, size and jitter, and the task for the observer was to decide which stimulus appeared more regular. To ensure that all stimulus conditions contained the same mean luminance, our texture elements were difference-of-Gaussian (DoG) micropatterns. We tested a variety of SF-distribution parameters from different filter response models to identify key factors that could be used for potential regularity read-out in the visual system.

## Results

To investigate how element spacing, size and jitter jointly affected perceived regularity, we generated 45 experimental conditions composed of three levels of element spacing, three levels of element size and five levels of jitter ranges (Fig 1). Five observers performed the paired comparison task, choosing on each trial the more regular-appearing stimulus (Fig 2). With 45 conditions, there were a total 1035 possible pairs (including same-condition pairs). The average responses to the 1035 pairwise comparisons are shown in a matrix format (Fig 3), separately for each observer (bottom right), for their group average (left), and for an ideal observer (upper right). Note this is like the Conjoint Proportions Plot of Ho et al. (2008) [18], but expanded to account for our 3-factor design. The pixel colors in the matrices indicate the percentage of trials on which stimulus $R_{ijk}$ (abscissa) of a pair is judged more regular than stimulus $R_{pqr}$ (ordinate). Note that pixels along the diagonal represent pairs having identical stimulus conditions, resulting in near-chance responses (green). For an individual subject's matrix, the pixel value can only be one of the five possible percentages (0%, 25%, 50%, 75% and 100%) based on four responses in each pair-wise condition, while the values in the group mean matrix can take more intermediate levels.

The overall pattern of the pairwise response matrixes are generally similar across observers: from the diagonal to the corner of the matrix (i.e. from smaller to larger jitter difference), the color gradually changes from green (50%) to yellow (100%), indicating the traditional influence of jitter level on perceived regularity as found in previous studies [5–7,13]. However, the

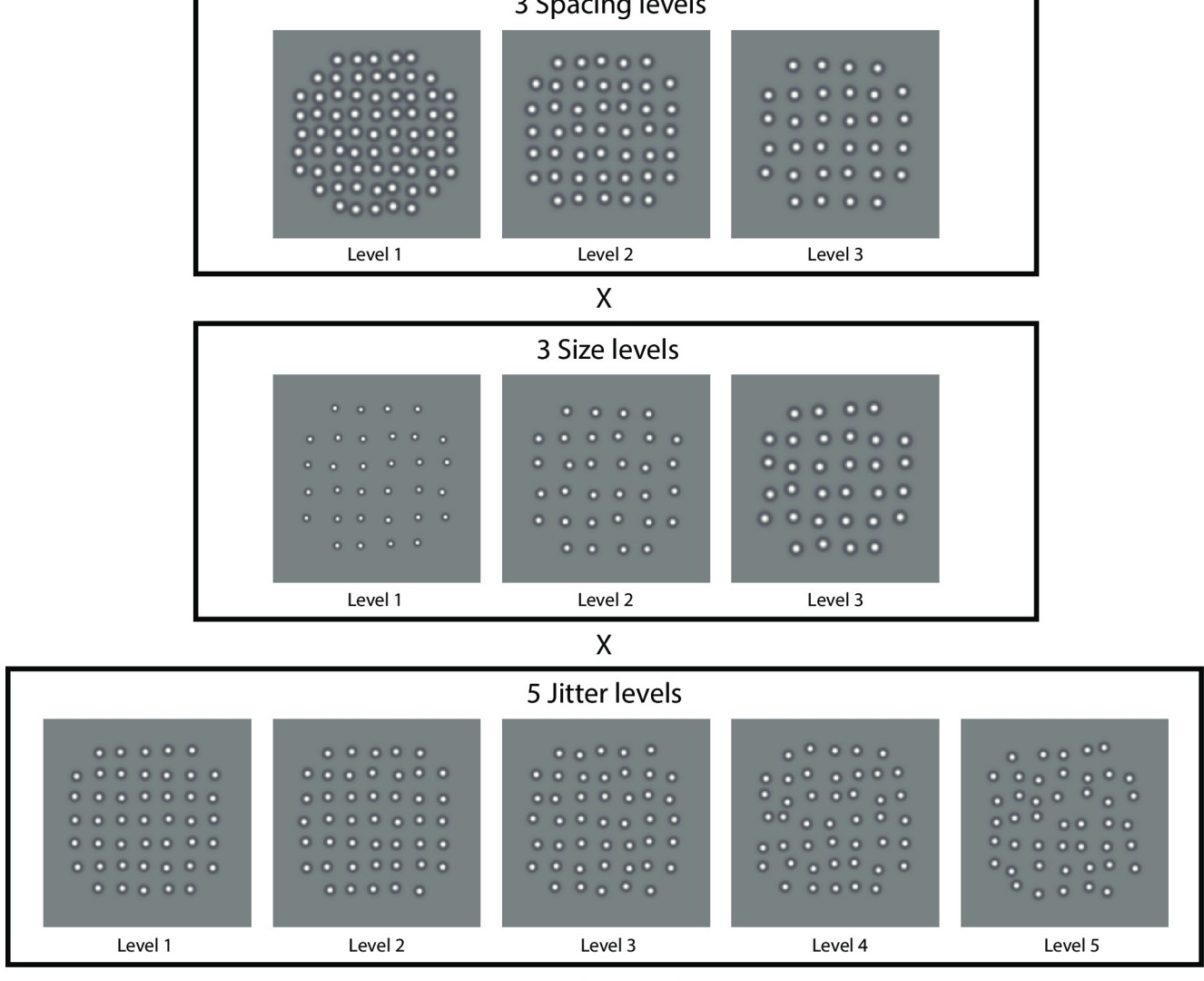

**Fig 1. Example stimulus texture patterns in the experimental design.** Textures were composed from DoG (Difference of Gaussian) micropatterns. Each stimulus texture was assigned one of three element spacing levels (top row), one of three element size levels (middle row), and one of five element jitter levels (bottom row), for a total $3 \times 3 \times 5 = 45$ conditions, and 1035 possible combinations of pairs $((44 \times 45)/2 + 45) = 1035$), including 45 same-condition pairs. Examples of 11 out of 45 conditions are depicted here. In the top row of stimuli, spacing level is 1 to 3 from left to right (small to large) at size level 3 and jitter level 2. In the middle row, both spacing level and jitter level are 3, and size level is 1 to 3 from left to right (small to large). In the bottom row, both spacing level and jitter level are 2, and jitter level is 1 to 5 from left to right (more regular to more irregular).

responses of the five observers are different from an ideal observer whose judgment is based solely on jitter alone (Fig 3 top right). In other words, the pixel color is not homogeneous within the big blocks (solid lines of the matrix, indicating jitter levels), suggesting appreciable effects of element spacing and size on perceived regularity by the human observers.

## 3-factor statistical model-based analysis of feature contributions

To evaluate the effects of element spacing and size statistically, and compare them with the traditional jitter effect, we first consider simple one-factor statistical models for element spacing

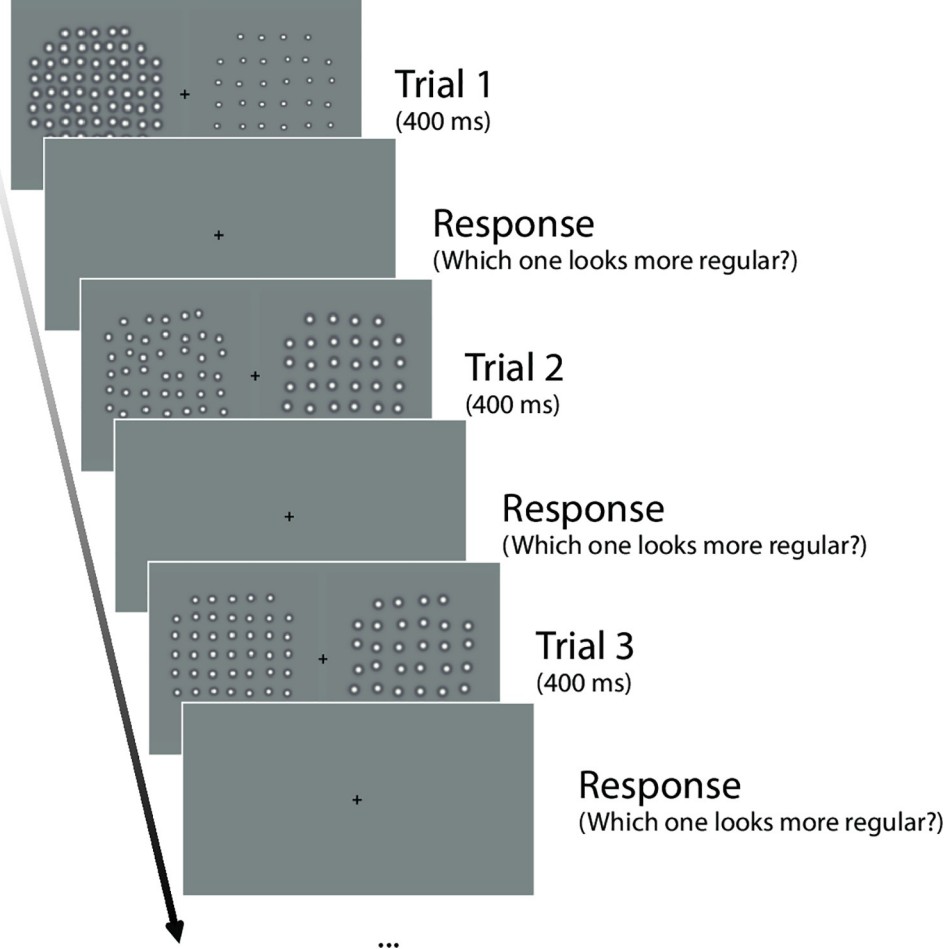

**Fig 2. Stimulus presentation protocol for three example trials of the pairwise psychophysical task.** A pair of element texture patterns was presented for 400 ms for each trial. Observers had unlimited time to choose the more regular-appearing pattern using a numeric keypad, followed by the next trial. A fixation target was shown throughout the experiment. Note that this schematic depiction is for demonstration purposes, and the actual size and scale of stimuli are as described in 'Design and Procedure'.

(Model 2), size (Model 3) or jitter (Model 4) separately, and fit them individually to the responses of each observer. These three models are tested against a baseline model (Model 1), which assumes no effects of element spacing, size and jitter, and therefore no difference in perceived regularity across the 45 experimental conditions. S1 Table shows the model comparison results for the five observers using likelihood ratio tests—see Methods. The results indicate that each of the three factors (element spacing, size and jitter) significantly contribute to regularity perception, and these results are highly consistent across observers. The effect of jitter is the strongest, which is on average 45 times stronger than the effect of size (S1 Table). The effect of spacing is, on average, about double that of size (S1 Table, 80.47/42.04 = 1.91). Note that, due to the significant two-way interactions (see below), the main effects described here should be interpreted with caution—we report them for clarity and to provide additional information. The subsequent models that include higher-order interactions provide a better description of how the variables influence perceived regularity.

To better understand how element spacing, size and jitter levels separately influence regularity, the parameter estimation of the one-factor models at each level of each factor are shown

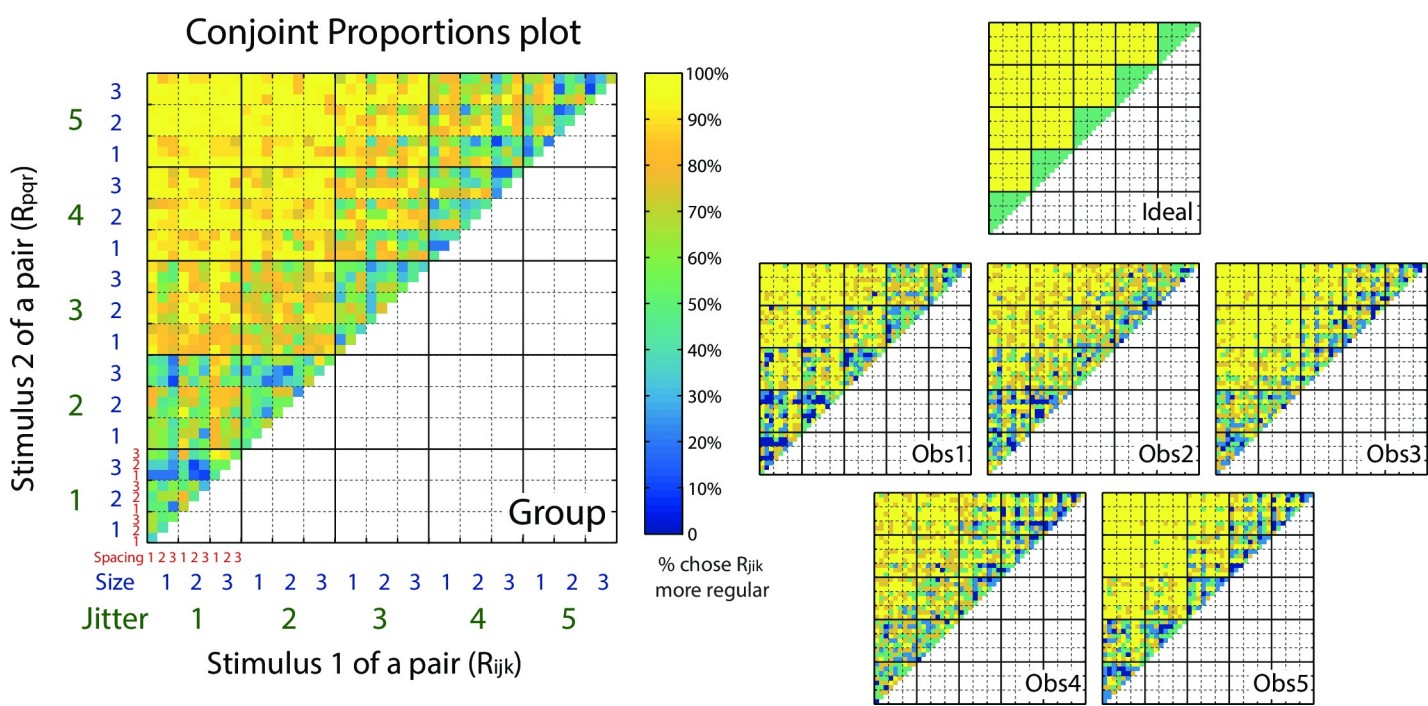

**Fig 3. Conjoint Proportions Plot.** Plots showing pairwise comparison results for regularity judgments, for an ideal observer (right top), the five individual observers (right bottom) and their group average (left). The color scale of the pixels in the matrices represents the percentage of trials on which stimulus $R_{ijk}$ (absicassa) is judged more regular than stimulus $R_{pqr}$ (ordinate). The data matrix for an ideal observer (right top) is plotted based on the difference of jitter only, and is uncontaminated by the effects of element spacing and element size. The group matrix (left) is calculated by averaging the response percentages across the five observers (right bottom). Element spacing is indicated by the smallest, red numerical labels 1–3. Element size is indicated by the blue numerical labels 1–3. The blocks in dotted lines represent different element size levels. Jitter level is indicated by the largest, green numerical labels 1–5. The blocks in solid lines represent different jitter levels.

for each observer in Fig 4. Jitter level (green) shows the strongest effect, with perceived regularity decreasing with larger amounts of jitter. This effect is very clear and consistent across observers, in line with the traditional definition and manipulation of texture regularity [5–7,13]. The effects of element spacing (red) and size (blue) are substantially smaller. Since the largest perceived regularity decrease is due to an increase in jitter, the group-average results were formed by first normalizing each observer's values to the decrease in regularity perception from the lowest to the highest jitter. We then averaged across observers for the group data (Fig 4 bottom right). Compared with the effect of jitter, for which the perceived regularity decrease is set to 100% on average at level 5 ($R^{\gamma}_5$), element spacing increased perceived regularity by 7% on average at level 3 ($R^{\alpha}_3$) and element size decreased perceived regularity by 7% on average at level 2 ($R^{\beta}_2$). This analysis provides a preliminary view of how the different variables influence regularity, which should be interpreted cautiously due to the presence of two-way interactions as explained below.

While the three factors all contribute to regularity judgment when analyzed separately, a further question is whether and how they interact and thereby contribute to regularity perception jointly. To this end, we firstly consider statistical models of all the possible two-way interactions between the three factors–the combinations of element spacing and jitter, element size and jitter, and element spacing and size. For each of the possible two-way interactions, we consider the contributions of two of the three factors (e.g. jitter and element spacing) to be either independent and additive (i.e. the two-additive-factors model) or dependent and nonlinear (i.e. the two-interactive-factors model). The two-interactive-factors model is then tested against the two-additive-factors model to evaluate whether the former predicts the data

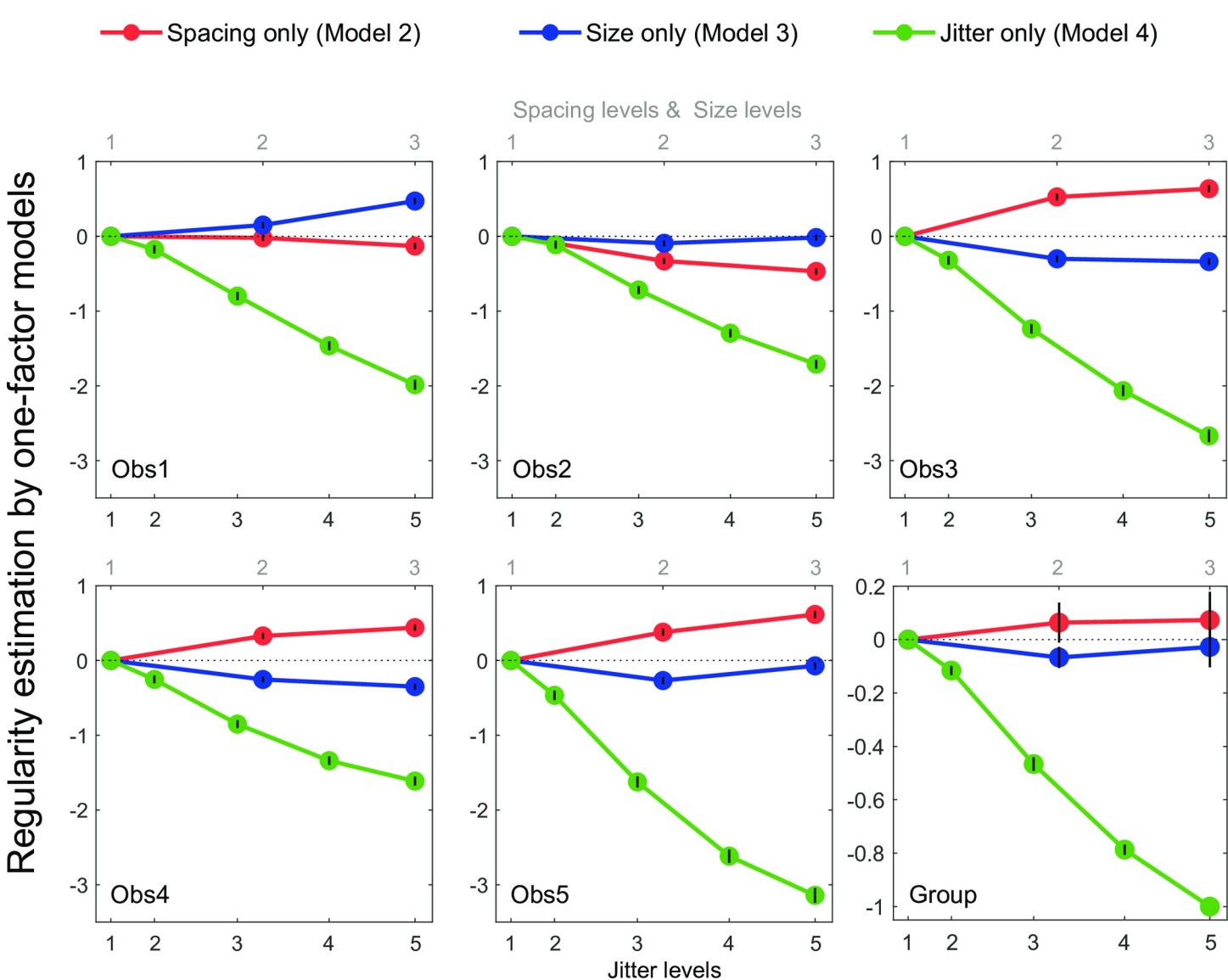

**Fig 4. Parameter estimates of the one-factor statistical models for each observer and their normalized group mean.** The estimated values are plotted as a function of element spacing level (red lines), element size level (blue lines), and jitter level (green lines). The three element spacing and element size levels correspond to the top scale in the abscissa, while the five jitter levels are indicated by the bottom scale in the abscissa, which correspond to the proportion of the averaged Pythagorean distance of the resulting element positions (see S7 Fig). For the individual graphs (Obs 1–5) the error bars represent standard errors of the parameter estimation from second partial derivatives of the log likelihood at the minimum [17]. For the group graph, the original parameters were normalized to the maximum absolute value, which is condition $R^{\gamma}_5$, for each observer, then averaged across observers. The error bars represent standard errors of the normalized parameters calculated across the five observers. No error bars for conditions $R^{\alpha}_1$, $R^{\beta}_1$ and $R^{\gamma}_1$ as they were set to 0 at each individual's level. Also no error bar is shown for condition $R^{\gamma}_5$ in group plot as it was set to -1 for normalization.

significantly better than the later. S2 Table shows the comparison results between these two nested models for the five observers. The results indicate that for both the element spacing × jitter and element size × jitter interactions, the interactive-factors models are significantly better than the additive-factors models across all observers (Model 5 vs. 8 and Model 6 vs. 9, with Bonferroni correction). These results suggest nonlinear, interactive contributions for element spacing × jitter and element size × jitter. The interaction effect of jitter with spacing is about double that for size (S2 Table, 177.69/76.21 = 2.33), consistent with the finding in the main effects (S1 Table) and supporting the general idea that spacing plays a more

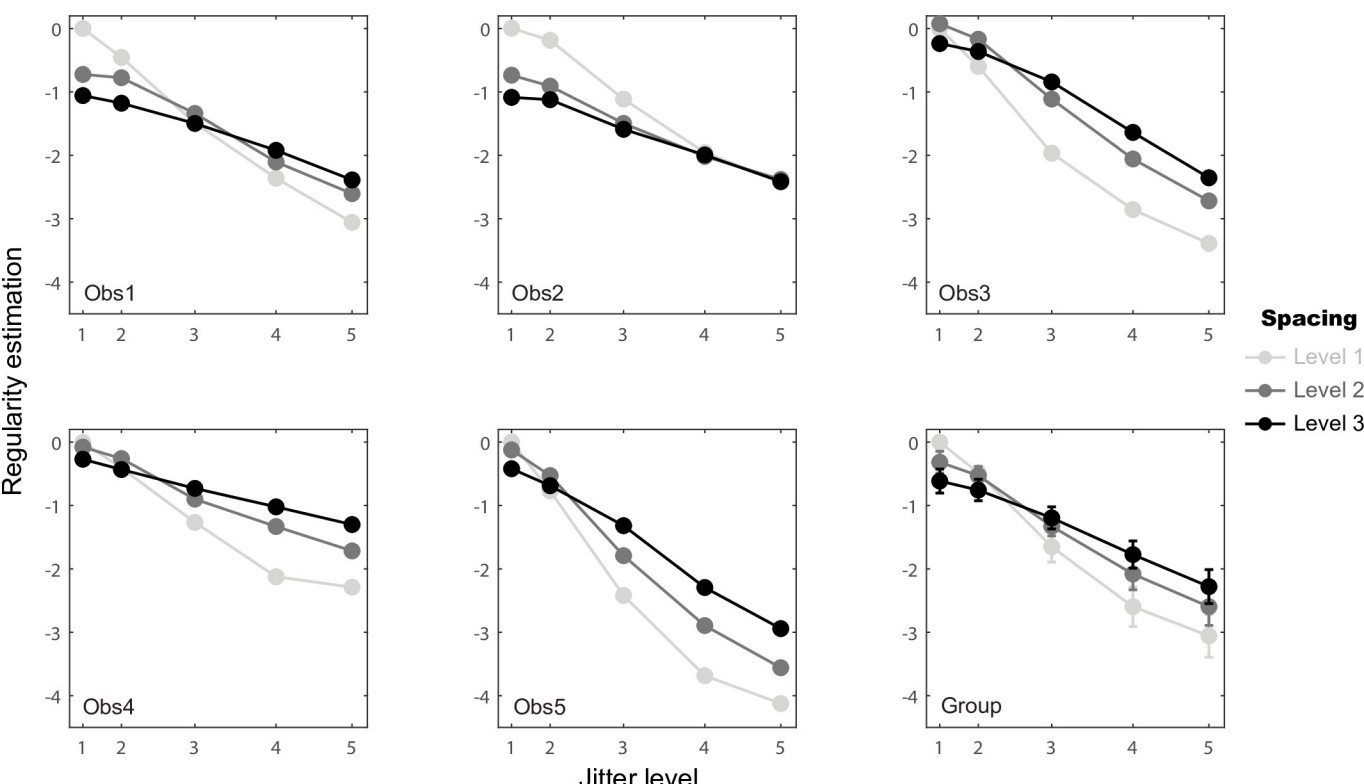

**Fig 5. Element spacing × jitter interaction based on the coefficients of statistical Model 8.** Estimated regularity values are plotted against jitter level (corresponding to the proportion of the averaged Pythagorean distance of the resulting element positions as in Fig 4) for the three element spacing levels separately (light to dark lines). Group data were calculated by pooling the individual data values together across the five observers and calculating their means and standard errors for each condition.

important role than size. For element spacing × size, the interactive-factors model is not significantly better than the additive-factors model (Model 7 vs. 10) and the deviance values are consistently very small across observers, suggesting the absence of a two-way interaction between element spacing and size on regularity.

To further understand the element spacing × jitter and element size × jitter interactions, we plotted the estimated regularity against jitter level for each element spacing level (Fig 5) and for each element size level (Fig 6). For the element spacing × jitter interaction, the jitter effect is stronger with smaller element spacing (lighter lines are steeper than darker lines in Fig 5). The mean jitter difference (level 1 vs. 5) is 1.87 times stronger for small element spacing comparing with large spacing (level 1 vs. 3) The element size × jitter interaction is relatively smaller, but the jitter effect is stronger with larger element size for all participants (darker lines are steeper than lighter lines in Fig 6). The mean jitter difference (level 1 vs. 5) is 1.51 times stronger for large element size comparing with small size (level 3 vs. 1) These results suggest that the visual system may utilize the distance between edges of elements when estimating pattern regularity (i.e. relative jitter) rather than between their centers (i.e. the absolute element positions) as the main index for regularity judgment.

In addition to the two-way interactions, we must evaluate the possibility of a three-way interaction effect, i.e. element spacing × size × jitter. This is an important step in three-way MLCM, which is not considered in the previous study involving three factors [24]. Considering two-way interactions only without checking the three-way interaction might result in misleading conclusions, since the two-way interactions may only be important at specific levels of

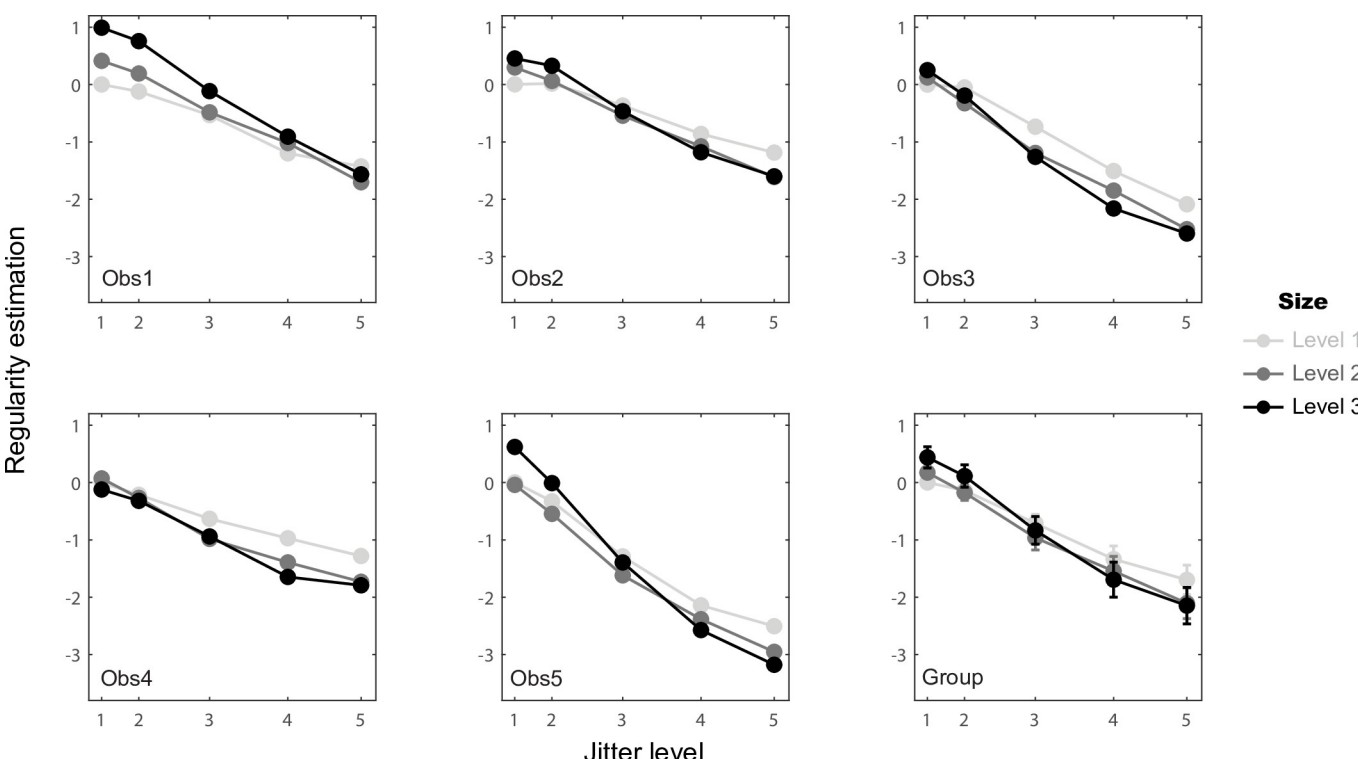

**Fig 6. Element size × jitter interaction from the coefficients of statistical Model 9.** Estimated regularity values are plotted against jitter level for the three element size levels separately (light to dark lines), otherwise as in Fig 5.

the third factor. To examine this idea, we consider the three-factors full model (Model 12, see Methods), which allows both two-way and three-way interactions, and the three-factors two-way interaction model (Model 11), which allows two-way but not three-way interactions. We then tested whether there is a significant three-way interaction by testing whether Model 12 can predict responses significantly better than Model 11 (using a likelihood ratio test between these two nested models). S3 Table shows these model comparison results for the five observers. The results show that the three-factors full model (Model 12) is not significantly better ($p > 0.01$) than Model 11 for most observers (four out of five) and the effect size (represented by deviance value) is generally small. This result suggests a relatively weak element spacing × size × jitter three-way interaction in regularity perception.

To summarise this series of statistical model comparisons, the effect strength (deviance) of all the main effects and interaction effects from S1–S3 Tables are depicted together in Fig 7. The effect of jitter is the strongest (the third bar in Fig 7), in line with the conventional manipulation of regularity using jitter [5–7,13]. Importantly, element spacing and size also contribute to regularity perception (the first and second bars in Fig 7), and each of them also interacts with jitter to contribute to perceived regularity jointly (the fourth and fifth bars in Fig 7). Element spacing more strongly influences regularity perception than element size (S1 Table), and its interaction with jitter is also stronger (S2 Table).

## Filter models for regularity encoding

In our earlier research on simultaneous regularity contrast, we proposed two types of image-computable models that could "read out" perceived regularity using descriptive parameters (e.g. peakedness, kurtosis, skew FWHM and standard deviation) of the distribution of image

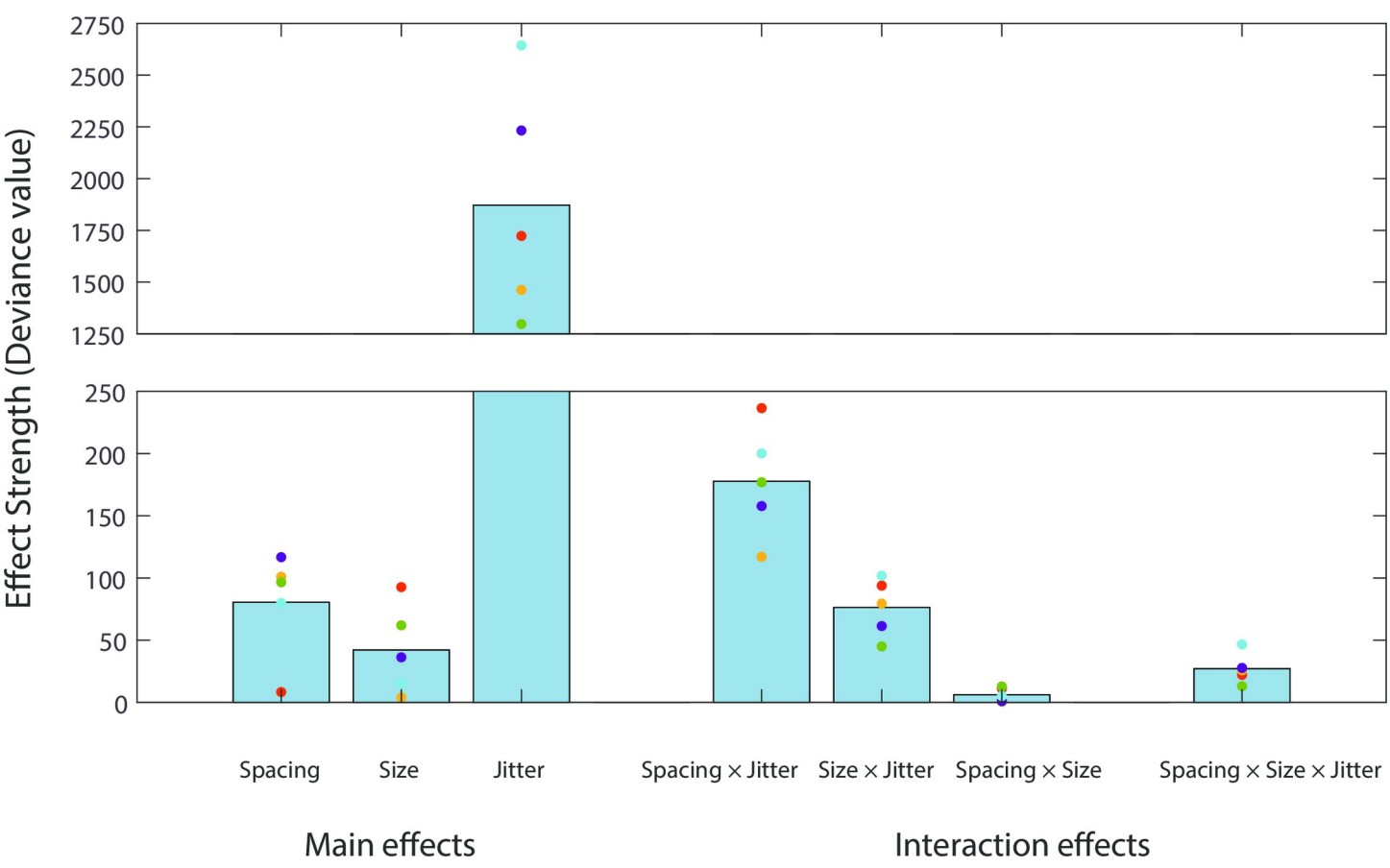

**Fig 7. Summary of the three-way MLCM statistical model comparisons.** The bars (from left to right) represent the mean effect strength (deviance) across the five observers for element spacing, size and jitter main effects (S1 Table) and element spacing × jitter, element size × jitter, element spacing × size (S2 Table) and three-way interactions (S3 Table). Colored circles represent individual data of five observers. The bar chart is depicted with a break between deviance values 250 and 1250, to improve visualization of small values.

responses from a bank of Gabor spatial filters that are selective for spatial frequency and orientation [7]. Here we reconsider these models, and also two other models, in light of the richer data from this study.

For all of these models, we first performed a similar Gabor wavelet analysis as described earlier [7], for the 45 experiment conditions used here (11 of the 45 sample images, each 256 × 256 pixels, are shown in Fig 1). Each image was convolved with a set of log Gabor filters ranging from low to high spatial frequencies (50 log-spaced SFs from 0.23 to 512 cycles per image), each at 12 orientations (0–165 degrees, with a 15-degree interval), using the LogGabor transform software customized from publicly available code [25]. For each input image the pixel values were initially squared to bring out the above-background contrast energy of the elements before filtering, somewhat similar to the concept of "second order" processing in the filter-rectify-filter (FRF) scheme [26]. The pixelwise root-mean-square (RMS) response (i.e. energy) of each filtered image was then calculated and normalized by the square of the Gabor filter size.

The size-normalized RMS values of each condition were then additionally normalized to equate total response across the 12 orientations (i.e. a mean normalization), so that the total

# SF model

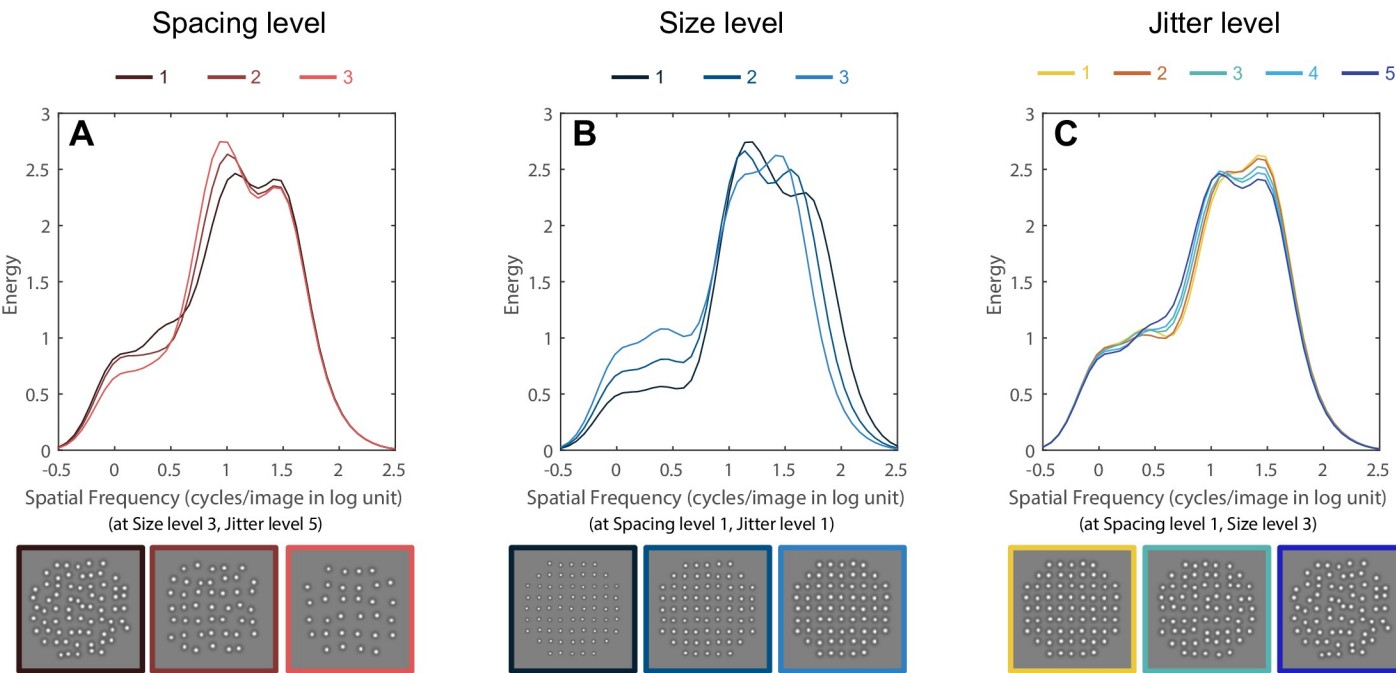

**Fig 8. Gabor wavelet analysis results for the SF-distribution model.** Each stimulus image from the experimental conditions (examples shown at the bottom) was filtered by a set of oriented Gabor filters (12 orientations, 0–165 degree, with a 15-degree interval) with a wide range of spatial frequencies (50 log-spaced SFs from 0.23 to 512 cycles per image). The pixelwise root-mean-square (RMS) responses (i.e. energy) of each filtered image was then calculated and normalized by the square of the filter size, and then averaged across orientations. The RMS responses are plotted as energy (ordinate) as functions of spatial frequency (abscissa). In (A), we plotted the analysis results for the three element spacing levels (from dark to light red lines) at size level 3 and jitter level 5 conditions, as other size and jitter combinations all show a similar pattern shift of the SF distribution with spacing–the highest peak shifts to the left. In (B), the analysis results for the three element size levels at spacing level 1 and jitter level 1 conditions are plotted with dark to light blue lines. Other spacing and jitter combinations also show a similar pattern shift of the SF distribution with size–the second highest peak shifts to the left. (C) depicts the SF distributions of five jitter levels (from yellow to blue lines) at spacing level 1 and size level 3 conditions. As shown in (C) and in our earlier study [7], higher regularities (i.e. lower jitter levels) make the SF distribution sharper while lower regularities (i.e. higher jitter levels) make the SF distribution flatter, though these effects are small for the stimuli used here. Other spacing and size combinations also give similar changes in the SF distribution.

amount of energy (and the mean energy) is the same across orientations and conditions:

$$NE_{(m,n)} = \frac{RE_{(m,n)}}{\frac{1}{12} \times \sum_{n=1}^{12}\left(RE_{(m,n)}\right)} \qquad \text{(Formula 1.1)}$$

where $NE$ is normalized energy and $RE$ is raw energy. $m$ indexes the 50 SF levels and $n$ indexes the 12 orientations.

**Regularity model based on the SF distribution.** First we consider a model which operates on the distribution of Gabor wavelet responses that is averaged across orientation [7]. To examine this "SF distribution" model, the $NE$ values from Formula 1.1 were averaged across the 12 orientations to yield the distribution of energy ($x$) as a function of spatial frequency (indexed by $m$):

$$x(m) = \frac{1}{12} \times \sum_{n=1}^{12}\left(NE_{(m,n)}\right) \qquad \text{(Formula 1.2)}$$

Fig 8 shows how the SF distributions $x$ change with element spacing, size and jitter. As shown in Fig 8A, B, changing element spacing and size changes the shape of the SF distribution. For the patterns used in this study the left high peak (~10 cpi) is associated with element

spacing and the rightmost high peak ($\sim 10^{1.5}$ cpi) is related to element size. The third, much smaller peak or shoulder at very low spatial frequencies (about log SF = 0, i.e. 1 cycle/image) reflects the shape of the circular aperture within which the elements were distributed in the experimental stimuli (Figs 1 and 2)—see *Discussion* for more details. The results show that increasing element spacing pushes the high peak to the left (Fig 8A), i.e. lower SFs, as seen earlier [16]. Increasing element size pushes the high peak to the right, toward higher SFs (Fig 8B). Note that since we applied mean normalization to each SF distribution (Formula 1.1), the shift of peakedness (i.e. maximum energy) is opposite to that of the previous predictions [16]. That is, the peakedness is increasing with larger spacing and smaller size rather than decreasing. This is because larger spacing and smaller size generate weaker overall filter responses, and after mean normalization the peaks become larger than in the conditions with larger elements and smaller spacing. The same SF-distributions without mean normalization are depicted in S1 Fig. The normalization procedure (Formula 1.1) is an important element of the model since the overall response (energy) values are equalized to facilitate a fair comparison across different conditions.

However the shape of the SF distribution changes only subtly with perceived regularity (Fig 8C)—it is slightly sharper and higher in peakedness at higher regularities (i.e. smaller jitter levels) and becomes flatter at lower regularities. This effect is consistent with our earlier findings [7], but the magnitude of the SF shape change as a function of jitter is much smaller here due to the more limited range of jitter used in these experiments—our earlier research [7] used a very wide range of jitter values (0–24.2 jitter min). The jitter range we tested here is smaller due to the restrictions of larger element sizes and smaller spacing values in this study

Thus the SF-distribution model would predict that element spacing and size have greater influence on regularity perception than jitter, due to the greater changes in shape of the SF-distribution. However, our data shows the opposite–the effects of spacing and size, which significant, are substantially less than the effect of jitter (Fig 7 and S1 Table).

**Regularity model based jointly on SF and orientation distribution.** The inadequacy of the SF-distribution model to explain our data led us to re-consider averaging over orientation, and retaining a richer bivariate representation, which we term here the "SF × orientation" distribution model. Although the same Gabor filter response analysis and normalization procedures were performed (see Formula 1.1), this model differs from the one above in that the Gabor filter responses were *not* averaged across the 12 orientations (i.e. omitting Formula 1.2). Fig 9 shows how the filter response distributions change with Gabor filter orientation (0–165 degree, with a 15-degree interval) for four example conditions. Fig 9A provides a standard "reference" condition with element spacing, size and jitter all at level 1. Since the stimulus texture is very regular, the high energy responses are concentrated around the grid orientations (vertical and horizontal), and also the four diagonal orientations due to the shape of the stimulus aperture and the mean normalization across orientations. The same response distributions across orientation without normalization are depicted in S2 Fig for comparison. Importantly, the energy concentrations are only modestly affected by element spacing and size changes (Fig 9B and 9C, respectively), while jitter substantially spreads out the energy concentrations (Fig 9D). Based on the SF × Orientation distribution model, perceived regularity is determined by the variance of the distribution of energy across orientation (i.e. high variance encodes high regularity) [7]. Thus this model would predict that jitter has the greatest influence on regularity perception since it spreads out the energy concentration, and thereby reducing the variance across orientation, while spacing and size have more limited influence. This prediction is in line with our data showing a strong jitter effect in regularity perception by human observers (Fig 7).

# SF × Orientation model

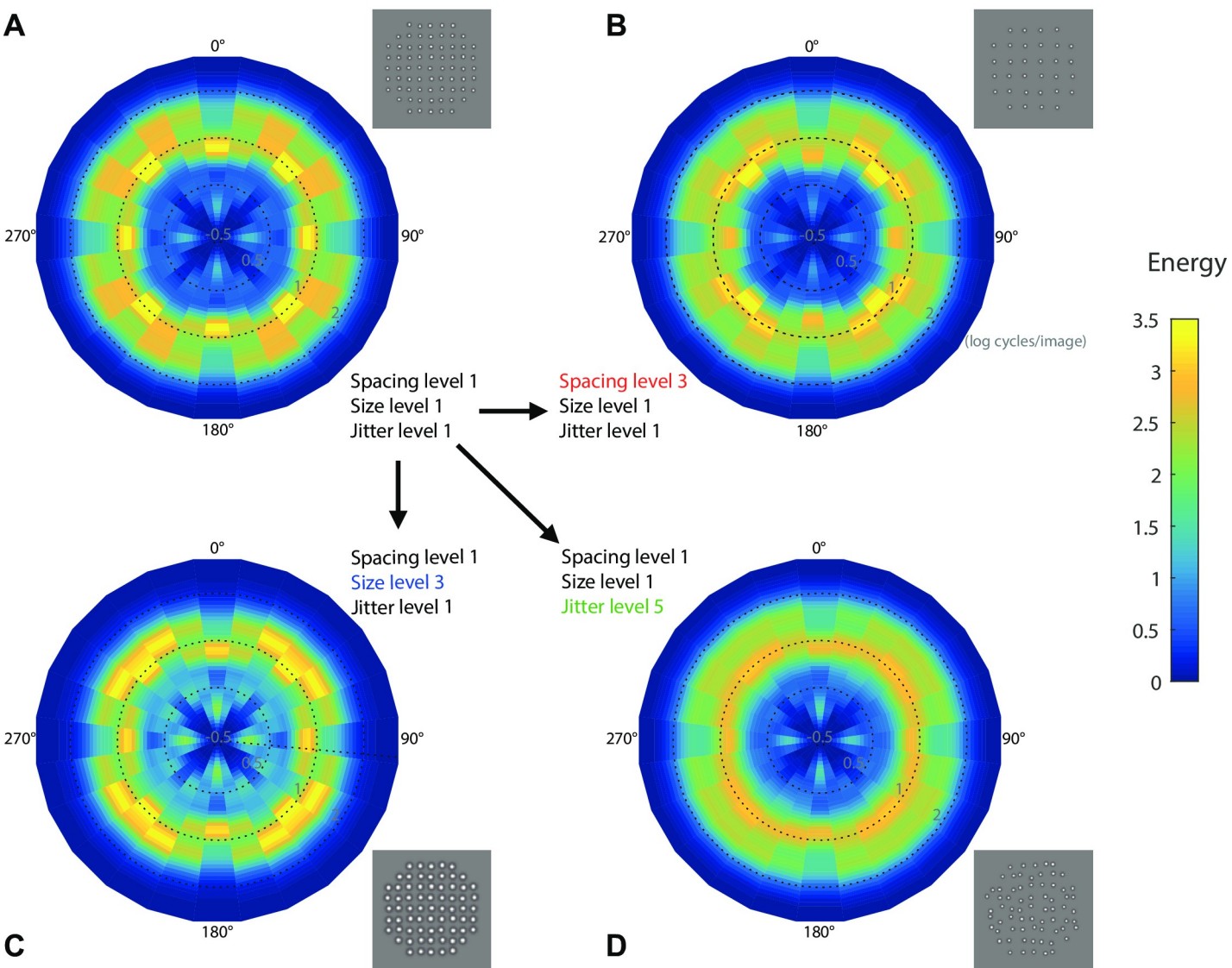

**Fig 9. Gabor wavelet analysis results for the SF × orientation distribution model.** The same SF analysis as in Fig 8 was performed, except the SF distributions were analyzed separately for each of the 12 orientations, and then the standard deviations of the distribution parameters across orientations were calculated [16]. The SF distributions are represented as polar image plots of energy (RMS responses) as a joint function of spatial frequency (distance from center, as scaled by circular dotted lines for three SF levels 0.5, 1 and 2 log cycles/image) and orientation (polar angle) for four selected experimental conditions. In (A) we show a standard reference condition for element spacing, size and jitter all at level 1, which can be compared to element spacing change (B), element size change (C) and jitter change (D). The results show that jitter spreads out the energy across orientations much more than spacing or size.

**Comparison of SF- and SF x orientation distribution models.** To compare how well the above two models can predict the human psychophysical results, we first calculated five parameters for the SF-distribution model–kurtosis, skew, full width at half maximum (FWHM), peakedness (maximum) and standard deviation of the averaged SF distribution across orientations (i.e. *x* in Formula 1.2) for each of the 45 conditions. The kurtosis (fourth

moment) of a distribution ($k$) of an experimental condition is:

$$k = \frac{E(x - \mu)^4}{\sigma^4} \qquad \text{(Formula 1.3)}$$

where $\mu$ is the mean of $x$, $\sigma$ is the standard deviation of $x$, and $E$ represents the expected value. Similarly, the skew of a distribution ($s$) captures the third moment of the distribution:

$$s = \frac{E(x - \mu)^3}{\sigma^3} \qquad \text{(Formula 1.4)}$$

The peakedness and standard deviation are measured as the maximum value and the standard deviation of the distribution $x$ respectively. FWHM measures the width of the distribution $x$ at half maximum.

For the SF × Orientation distribution model, we first calculated these parameters for each orientation separately, and then obtained the standard deviation of each parameter across the 12 orientations. The SF-distribution at an orientation $n$ is defined as $x_n$:

$$x_n = NE_{(m,n)} \qquad \text{(Formula 2.1)}$$

We calculate kurtosis for $x_n$ the same way as in Formula 1.3:

$$k_n = \frac{E(x_n - \mu)^4}{\sigma^4} \qquad \text{(Formula 2.2)}$$

The standard deviation of the 12 kurtosis values ($k_{1-12}$) is calculated as:

$$k_{std} = \sqrt{\left(\frac{1}{12 - 1} \times \sum_{n=1}^{12}(|k_n - \mu|^2)\right)} \qquad \text{(Formula 2.3)}$$

where $\mu$ is the mean of $k_n$.

We calculate skew for $x_n$ the same way as in Formula 1.4:

$$s_n = \frac{E(x_n - \mu)^3}{\sigma^3} \qquad \text{(Formula 2.4)}$$

The standard deviation of the 12 skew values ($s_{1-12}$) is calculated as:

$$s_{std} = \sqrt{\left(\frac{1}{12 - 1} \times \sum_{n=1}^{12}(|s_n - \mu|^2)\right)} \qquad \text{(Formula 2.5)}$$

where $\mu$ is the mean of $s_n$. The other three parameters (i.e. FWHM, peakedness and standard deviation) are calculated using the same framework for the SF × orientation distribution model.

We then separately correlated the resulting 10 parameters, five from the SF-distribution model and five from the SF × Orientation distribution model, with the regularity estimation of the 45 experimental conditions from the full model (i.e. Model 12, which explains the most deviance—see Discussion and S3 Fig) averaged across the five observers. We use the regularity estimation from Model 12 rather than Model 11 here because Model 12 is more straightforward and explains more deviance than Model 11 (S3 Table, three-way interaction—also see Discussion). Since we have a large number of trials for each image, we consider it better to reduce the bias rather than the variance of the model estimates. Introducing 16 additional parameters (Model 11 vs. Model 12; see Methods) should lead to better estimates without much overfitting. The five parameters from the SF × Orientation distribution model (Fig 10B, averaged r = 0.71) show much stronger correlation with perceived regularity than the five

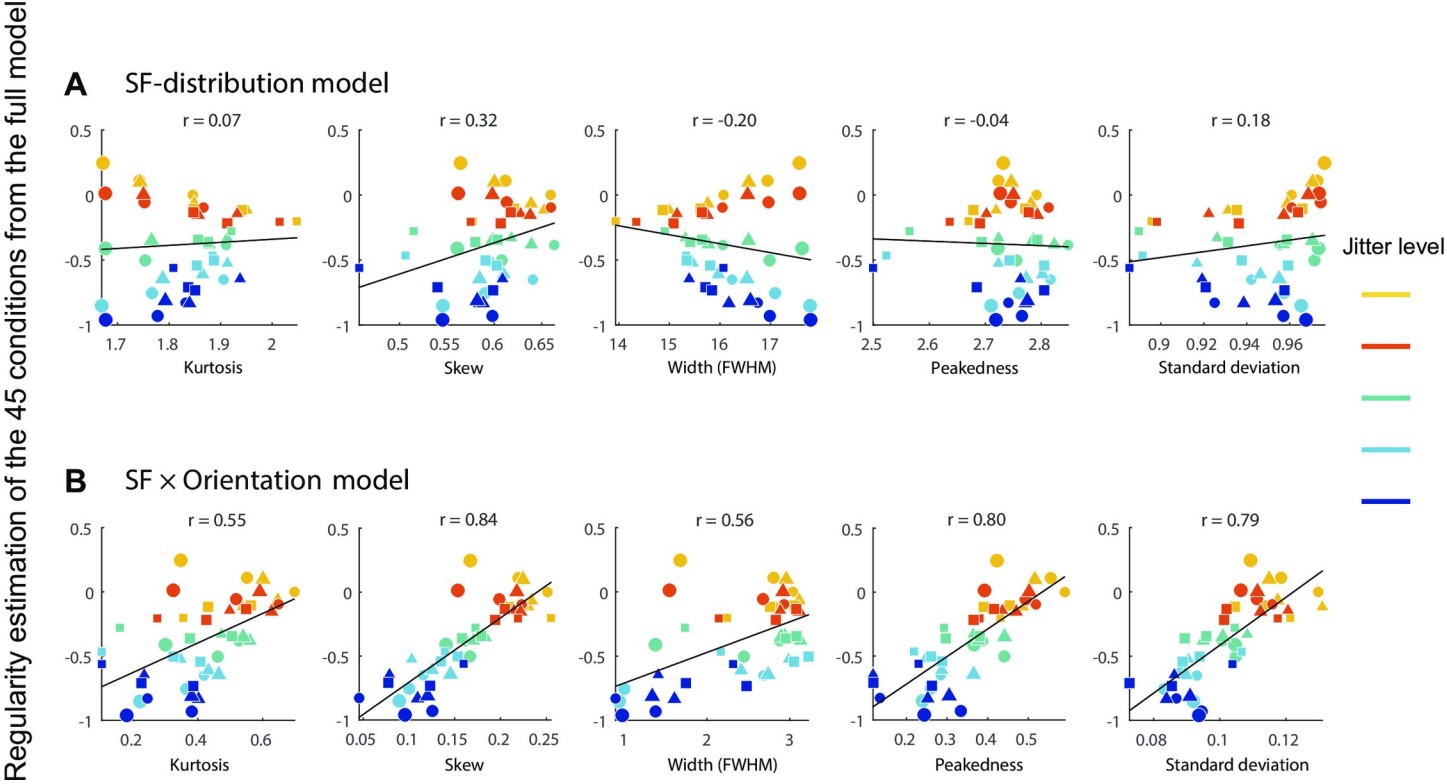

**Fig 10. Comparison of SF-distribution model and SF × orientation distribution model in predicting psychophysical results.** Perceived regularity estimates for the 45 experimental conditions from the full statistical Model 12 (see S3 Fig) are correlated with five SF-distribution parameters (kurtosis, skew, FWHM, peakedness and standard deviation) from SF-distribution model (A) and another five parameters from SF × orientation distribution model (B). Color codes for the five jitter levels (yellow to blue). The marker size (small to large) corresponds to element size (level 1 to 3), and the marker shape (circle, triangle and square) corresponds to element spacing (level 1 to 3). The SF × orientation distribution model shows stronger correlations with perceived regularity, across all of the five parameters.

parameters from the SF-distribution model (Fig 10A, averaged r = 0.16), supporting the SF × Orientation distribution model as a better model for regularity encoding.

To identify which parameters are important in regularity perception, we further test which of the 10 SF-distribution parameters can best predict regularity perception in these experiments, we performed a stepwise regression analysis (see *Methods* for details) to find which parameters can best predict regularity independently from their correlations with the other parameters. This procedure resulted in only one parameter being included in the statistical model, which is the skew from the SF × Orientation distribution model (i.e. the variance of the SF-distribution skew across 12 orientations). Skew is the parameter that is most correlated with regularity (Fig 10B) and explains 70% of the variability of estimated regularity from the full model (r = 0.839, $R^2$ = 0.704, adjusted $R^2$ = 0.697, p < .001), which is generally considered strong. Because other parameters such as peak height and standard deviation are also strongly correlated with regularity (Fig 10B), this result suggests these parameters might be too correlated between each other (collinearity) for a statistical model to distinguish how they might separately contribute to regularity.

**Regularity models based on the orientation distribution.** Given that both the above models ultimately utilized the SF-distribution for regularity encoding, it would seem prudent to consider analogous models which use the orientation-distribution as the basis for regularity encoding, as suggested by Mark Georgeson (personal communication). To this end, we swapped SF with orientation at the outset, i.e. the mean normalization procedure (see Formula

1.1 for details):

$$NE_{(m,n)} = \frac{RE_{(m,n)}}{\frac{1}{50} \times \sum_{m=1}^{50}(RE_{(m,n)})}$$ (Formula 3.1)

We normalized this raw energy to equate total response across the 50 SF levels so that total amount of energy (and the mean energy) is the same across SF and conditions.

For the "Orientation-distribution" model, *NE*s were then averaged across the 50 SF levels to give the energy distribution as a function of orientation:

$$x(n) = \frac{1}{50} \times \sum_{m=1}^{50}(NE_{(m,n)})$$ (Formula 3.2)

S4 Fig shows the orientation distributions (*x*) as functions of element spacing, size and jitter. The two energy peaks in each plot are concentrated at orientations 0 and 90 degrees, reflecting the nature of the grid pattern stimuli. However, the shape of the orientation distribution changes only very subtly with any of the three factors. Consequently, we did not further consider this model.

For the Orientation × SF distribution model, we calculated the four orientation distribution parameters for each SF separately, and then the standard deviation of each parameter was calculated across the 50 SFs. The orientation-distribution at a SF level *m* can be defined as $x_m$:

$$x_m = NE_{(m,n)}$$ (Formula 4.1)

The normalized energy (*NE*) here is from Formula 3.1, which is different from the *NE* in Formula 1.1. Then we calculated kurtosis for $x_m$ the same way as in Formula 2.2:

$$k_m = \frac{E(x_m - \mu)^4}{\sigma^4}$$ (Formula 4.2)

Then the standard deviation of the 50 kurtosis values ($k_{1-50}$) was calculated as:

$$k_{std} = \sqrt{(\frac{1}{50-1} \times \sum_{m=1}^{50}(|k_m - \mu|^2))}$$ (Formula 4.3)

where $\mu$ is the mean of $k_m$. The other three distribution parameters (i.e. skew, peakedness and standard deviation) were calculated in a similar manner as for the SF × Orientation distribution model.

Similar to the SF-distribution and SF × Orientation distribution models comparison, we calculated the same distribution parameters for the Orientation-distribution model and Orientation × SF distribution model–kurtosis, skew, peakedness and standard deviation of the averaged orientation distribution across SF for each of the 45 conditions (see Formulas 1.3 and 1.4 for example). We omitted FWHM due to the shape of orientation distribution, which has two major peaks and sharp edges, which are not suitable for FWHM estimation.

The four parameters from the Orientation-distribution model show lower correlation (r = 0.28, 0.30, 0.74, 0.79 for kurtosis, skew, peakedness and standard deviation respectively) with perceived regularity of our data than the four parameters from the SF × Orientation distribution model (Fig 10B) but higher than those from SF-distribution model (Fig 10A). The four parameters from the Orientation × SF distribution model show the lowest correlation with our regularity data compared with the other three models (r = 0.04, -0.27, -0.23, -0.29 for kurtosis, skew, peakedness and standard deviation respectively). These results again indicate that the Ori × SF distribution model should not be considered further.

**Summary of regularity encoding models.** Our best performing model is the SF × Orientation model, i.e. the variance of the SF-distribution skew across orientation. We suggest this metric as the basis of a potential "read-out" of perceived regularity.

## Discussion

We have demonstrated a nonlinear influence of element spacing, size and jitter on texture regularity perception. This is the first research using three-factor Maximum Likelihood Conjoint Measurement (MLCM) to estimate how three physical dimensions jointly contribute to a perceptual judgment. We utilized a novel statistical approach to evaluate not only how two-way but also three-way interactions of the variables influence regularity. In addition to the well-known effect of positional jitter on perceived regularity, we also found effects of element spacing and size, in which the former is stronger than the latter. These results suggest that the visual system may utilize the distance between edges of elements as a reference for regularity judgment rather than using the distances between centers of elements.

### Implications for the use of MLCM

The main goal for the traditional MLCM approach has been to determine whether two or more factors contribute to a perceptual judgment, with or without any interactions. However, in a three-factor design, such an approach could miss potentially important two-way and three-way interactions. Although the two-way interactions can be tested by applying the original MLCM procedure to every pair of the three factors [24], we cannot use this method to test whether the three-way interaction is significant. Here we extend 2-factor MLCM and provide a new statistical framework to study all three variables simultaneously, allowing us to split the deviance explained by the full model into one-way, two-way and three-way effects, and thereby obtain a more full understanding of how different variables interact with each other.

In the present study we systematically examined the main effect of each factor (Models 2 vs. 1, Models 3 vs. 1, Models 4 vs. 1), two-way interactions (Models 5 vs. 8, Models 6 vs. 9, Models 7 vs. 10) and the three-way interaction (Models 11 vs. 12). This three-step framework provides a robust and efficient tool to investigate different kinds of perceptual processing, particularly the interactions for multiple-factor designs.

### Proposed model of perceived regularity

Our results provide theoretical implications about regularity encoding in the visual system. We found that our SF × Orientation model, in which regularity was encoded by the variance in SF skew across orientation accounted for 70% of the variance in regularity perception. This is consistent with other studies showing that there are neural mechanisms sensitive to skewed statistics, for example, the skewness of the pixel luminance histogram is correlated with the perception of surface gloss [27]. We argue that the variance of the spatial frequency wavelet response distribution across orientation may provide a perceptual read-out for regularity.

In the examination of potential regularity encoding models we applied an initial squaring of image pixel intensities, which is different from our earlier approach [7]. This provides a rudimentary "second-order" processing that gives the third highest peak in the SF response histogram, at low SFs (at about 1 cycle/image), which carries the information about the circular area in which the DoG elements are distributed in the experimental stimuli. Without this early squaring, the effect of jitter in changing SF-distribution shape is less clear (see S5 Fig), and the correlations between SF-distribution parameters and perceived regularity would also be reduced greatly (i.e. in Fig 10 the averaged r would become 0.06 for the SF-distribution model and 0.33 for the SF × orientation distribution model). This result may suggest that the

element alignment near the edge of the aperture within which texture elements are distributed might affect regularity judgments—this idea is a potential line of future investigation.

## Applying the SF × orientation distribution model to natural images

Naturalistic stimuli have been used in many visual perception studies. Nevertheless the majority of studies on texture perception continue to use simpler laboratory stimuli, for the simple reason that they afford the opportunity to manipulate stimulus parameters in a well-defined way, and in the case of psychophysics, to design behavioural tasks that involve judgments of salient perceptual dimensions, such as regularity. Only a handful of recent papers have explored the topic of texture regularity [5,6,16]. All these studies have employed lattice patterns, for the sensible reason that for a topic in its infancy it is best to begin with stimuli that are relatively straightforward to parametrize and manipulate. In this study the use of lattice patterns has enabled us to address the important and previously unanswered question of whether other texture parameters, such as element spacing and element size, and their interaction with element positional "jitter", can affect regularity perception.

To demonstrate that our results apply to more than simple lattices of texture elements, we further test our preferred SF × Orientation model against three natural images with distinctly different perceived levels of texture regularity. In this model, irregularity has the effect of spreading out the stimulus energy in different orientation bands, thereby reducing the variance in energy across orientation (S6 Fig). As explained earlier there are a number of candidate parameters that capture this spreading effect and we have calculated five of them, as described in Formulas 2.1–2.5 and Fig 10B. Their values are (from more regular to less regular natural images): Kurtosis$_{std}$: 0.59, 0.1527, 0.07; Skew$_{std}$: 0.27, 0.25, 0.19; FWHM$_{std}$: 2.09, 3.73, 1.74; Peakedness$_{std}$: 0.53, 0.20, 0.18; Standard deviation$_{std}$: 0.12, 0.10, 0.08. Four of the five parameters (kurtosis, skew, peakedness and standard deviation) show a correlation with perceived regularity. These findings are in accordance with the results from our lattice stimuli (Figs 9 and 10B), suggesting ecological validity for our model. A possible future research direction could be to improve the model by more systematically applying it to naturalistic images and/or synthesized complex images.

## Potential limitations of 3-factor MLCM and associated statistical models

In theory, the manipulations of the three factors we examined here (element spacing, size and jitter) are independent of one another. Assessing the validity of this assumption for the stimuli used in this study is important, in case of unexpected artifacts that might influence stimulus sampling (e.g. larger element size and smaller spacing may reduce the freedom of the jitter position of element). To this end, we calculated the actual element positions (Pythagorean distance relative to the grid) from 184 sample images for each condition and plotted the results of the most different conditions in S7 Fig. This plot shows that the true element positions relative to the grid for the five levels gradually decreased with jitter range (e.g. deviation below the diagonal in S7A and S7B Fig). In other words, the actual jitter positions of the stimuli are slightly smaller than the jitter ranges we specified for the five jitter levels, especially for higher jitter levels. A potential explanation lies in the procedure we used to prevent overlap of elements, which may reduce the freedom of element placements, especially in high jitter conditions. In other words, elements that would otherwise overlap were reassigned to other jitter locations within the same range [7]—as a result, the final element positions were restricted to locations that would not result in overlap with other elements. Note that this systematic difference in jitter level should not affect our MLCM model-fitting results, because we used non-parametric models. In other words, no specific parametric form of the variation of the

estimations with magnitude along the three-dimensions (i.e. element spacing, size and jitter levels) was imposed upon the data [17]. Importantly, the actual jitter produced in the stimulus images is not different between different element spacing levels (S7A Fig, dark line vs. grey line) and element size levels (S7B Fig, dark line vs. grey line). In other words, the manipulation of jitter is largely independent from the manipulation of element spacing and size. To avoid the small difference between the jitter ranges we defined the actual jitter positions of stimulus elements from affecting our data representation, all the plots with the five jitter levels (e.g. Figs 4–6) were based on the final Pythagorean distance of the element positions (i.e. the ordinate values in S7 Fig).

To examine the validity of our statistical model fitting procedure, we compared the parameter estimation of the full model against the real data of a naive participant (Observer 4) and simulated responses. The responses for all the possible 1035 pairs of the 45 experiment conditions were simulated 400 times each, based on a normal distribution for each condition (mean of each of the 45 conditions is set to the same as the 45 parameter estimates of the full model from real data).

The standard deviations (SDs) of the normal distributions were equated across the 45 conditions, as part of our "full model" assumptions (see Model 12). As shown in S8 Fig, with the fixed SDs, the parameter estimation of the full model from real and simulated responses are perfectly matched (the dark dots are aligned with the diagonal), confirming that the model-fitting procedure is correct and unbiased. However, if this intrinsic assumption is violated (e.g. different internal noise levels across conditions leading to different SDs of the normal distributions), the parameter estimation from simulated responses showed a slight distortion, deviating systematically from the diagonal (light dots in S8 Fig).

This result suggests that if our model assumption (i.e. the SDs are the same across conditions) is incorrect, the estimated parameters we reported might be influenced by artefacts at some level. In other words, if the actual amount of internal noise differs across conditions, the estimated parameters might be slightly biased (i.e. "varied SD" results in S8 Fig). In this study, it is not clear whether the internal noise changes with jitter level (and/or with element spacing and size levels). If the internal noise is approximately uniform across the 45 conditions, our model-fitting procedure should work accurately.

## Conclusion

This study has revealed a nonlinear relationship between effects of element spacing, size and jitter position in contributing to perception of texture regularity. Our results support the idea that the visual system may utilize the variance across orientation of parameters such as the skew of the spatial-frequency distribution of wavelet responses as a code for perceived regularity.

## Methods

### Ethics statement

The experimental protocols were approved by the Research Ethics Board (REB) of the Research Institute of McGill University Health Center (RI-MUHC) and conducted in accordance with their guidelines. All participants gave written informed consent before taking part in the experiment.

### Apparatus and stimuli

Stimuli were presented on a CRT monitor (Sony Trinitron GDM-F520, 20 inches, 1600 × 1200 pixels, 85 Hz) with a viewing distance 85 cm. The stimulus patterns were textures consisting of

difference of Gaussian (DoG) elements with light centers / dark surrounds and isotropic orientation (zero-balanced, standard deviation sigma = 2.43 arcmin, minimum luminance 7 cd/m$^2$, maximum luminance 115 cd/m$^2$), which were generated using the Matlab function 'CreateProceduralGaussBlob' in Psychophysics Toolbox [28–30] and presented on a mid-grey background (61 cd/m$^2$). Luminance was measured with an Optikon universal photometer (Optikon Corp. Ltd., Ontario, Canada), and linearized using Mcalibrator2 [31].

Texture stimuli consisted of DoG micropatterns placed at locations randomly perturbed ("jittered") with respect to a notional grid. Each stimulus texture was assigned one of five jitter levels, one of three element spacing levels, and one of three element size levels. Therefore, there were 5 × 3 × 3 = 45 conditions in total. Examples of 11 stimuli are depicted in Fig 1. Each trial consisted of two stimuli (a pair) from the 45 conditions, for a total 1035 possible pairs with an exhaustive approach [17]. Each stimulus texture was presented within a circular area, 2.91 deg in diameter. The two stimuli of a pair were presented 1.95 deg apart, to the left and right of a center fixation cross (Fig 2). The two stimulus patterns of a pair were also subjected to a small vertical jitter (within 4.87 arcmin up or down) independently, to avoid alignment of the two patterns within a trial and to avoid potential afterimages across trials, which might otherwise occur when the texture is highly regular [7].

Elements were placed on a 9 × 9 notional grid (inter-element distance 17.52 arcmin) for the element spacing level 1 condition. For spacing levels 2 and 3 the notional grids were 7 × 7 and 6 × 6 (interelement distances 21.41 and 25.30 arcmin respectively). The diameters of each element were 9.73, 12.65, 15.57 arcmin for element size levels 1 to 3 respectively.

Each element was randomly and independently jittered in its horizontal and vertical positions, with respect to its position on the aforementioned notional grid. The elements of each pattern for a given condition were assigned to one of the five jitter range levels (uniformly distributed): 2.92, 3.89, 5.84, 7.79 and 9.73 jitter arcmin. The Pythagorean distance ranges for the five levels are 1.12, 1.49, 2.23, 2.98 and 3.72 arcmin (see the abscissa in S7 Fig). The average Pythagorean distance of the resulting element positions relative to the grid for the five levels are 1.12, 1.45, 2.07, 2.76 and 3.40 jitter arcmin (see the ordinate in S7 Fig). Importantly, the manipulation of jitter is generally independent from the manipulation of element spacing and size (see Discussion).

### Design and procedure

Five observers with normal or adjusted-to-normal visual acuity participated in the experiment, including four naïve observers (Observers 2, 3, 4 & 5) and one of the authors (Observer 1).

Each observer completed 4140 trials (1035 pair judgments × 4 repetitions) in 20 blocks, each containing 207 trials. Each pair was presented in a random order and the conditions on successive trials were unpredictable. In each trial the positions of two paired stimuli were also randomly assigned to the left and right.

In each trial a pair was presented for 400 ms followed by a response period unrestricted in duration (Fig 2). Observers pressed buttons on a numeric keypad to indicate which stimulus of a pair appeared to have a more regular pattern. Each succeeding trial started 500 ms after the response to the preceding one. A fixation cross was always presented in the middle of the screen, and observers were instructed to fixate and not to make eye movements during stimulus presentations.

### Data analysis

**3-factors MLCM.**   Responses to the 1035 image pairs are first summed across the four repetitions for each observer. We then attempt to predict a participant's responses as a function of

the element spacing, element size and jitter level on each of the image pairs. This analysis is performed individually on the data from each participant, and results in a total of six factor variables: three for the first stimulus and another three for the second stimulus of a pair (Fig 3 left). We reduce the number of factor variables by half by using an approach similar to the Maximum Likelihood Conjoint Measurement (MLCM) method proposed by Knoblauch and Maloney [17]. This method assumes the factor variables from the first and second images to be the exact opposite of each other, which is equivalent to estimating the perceived regularity of each image. To understand how the three factors–element spacing, size and jitter–influence perceived regularity, we compare a series of models to examine their contributions separately and jointly. To assess how each variable directly influences regularity (i.e. main effects), we test whether the models that include only one of the three variables (Models 2 to 4) perform better than chance. To test whether two of the three variables interact to influence regularity (i.e. two-way interactions), we contrast models that include the additive influence of two of the variables (Models 5 to 7) with another model that assumes the same two variables interact (Models 8 to 10). To test whether the three variables interact and influence regularity jointly (i.e. three-way interaction), we compare a model that incorporates all three two-way interactions (Model 11) with the full model (Model 12), which estimates regularity separately for each image.

This series of likelihood ratio tests between two nested models is inspired by how three-way ANOVAs are performed, but it is applied here to results from a two-alternative forced choice paradigm.

For each of the aforementioned comparisons between models, we calculate the number of correctly predicted responses for each model, which follows a binomial distribution. This number can be expressed as a likelihood, i.e. a ratio of how many responses are predicted correctly to the total number of responses. The deviance is twice the difference between the log-likelihood ratios of two nested models, which asymptotically follows a Chi-Square distribution as the sample size increases [32]. We use this statistic to test whether one model performs significantly better than the other. These likelihood-ratio tests were performed in the R programming language (R Development Core Team, Vienna, Austria) using the 'anova' function, where we specify the distribution to be Chi-square. The model fitting was also done in R using its Generalized Linear Model (GLM) function.

**Three-way MLCM statistical models (Model 1 to 12).** The goal of each of the statistical models is to predict the decisions of each observer as to which image is more regular. We assume this decision is based on the difference in perceived regularity, $\Delta$, between the first and second stimuli, $R_{ijk}$ and $R_{pqr}$:

$$\Delta = R_{ijk} - R_{pqr} + \varepsilon$$

where $\varepsilon$ represents the internal noise in the observer's decision process. Positive values of $\Delta$ lead to the model choosing $R_{ijk}$ as more regular, while negative values lead to the model choosing $R_{pqr}$ as more regular.

A conventional probit transformation of $\Delta$ is used [17] to predict the response $\hat{y}$, which ranges continuously from 0 to 1.

$$\hat{y} = probit(\Delta)$$

The regularity perception of each image ($R_{ijk}$ and $R_{pqr}$) within each pair is separately estimated by a total of 12 models with different parameters. Model 1 is the baseline model and estimates regularity to be the same for all images. Models 2 to 4 estimate regularity from one of the three main parameters (spacing, size or jitter). Models 5 to 7 estimate regularity from pairs

of the three main parameters, assuming they separately influence regularity. Models 8 to 10 also estimate regularity from pairs of the three main parameters, but assume the two parameters interact with each other to influence regularity. Model 11 is a combination of models 8 to 10 which includes all three two-way interactions. Model 12 is the full model, which assumes a three-way interaction and estimates regularity separately for each image.

**Baseline model.** The simplest model only has one parameter and predicts regularity to be the same for all images:

$$R_{ijk} = 0 \tag{Model 1}$$

We compare it to each of the one-factor models to evaluate the influence of single factors on regularity.

### Additive models (Models 2 to 7). One-factor models

This type of model attempts to predict the regularity of an image based on only one of the three factors. It assumes that either only element spacing ($\alpha$, level i = 1–3), only element size ($\beta$, level j = 1–3), or only jitter level ($\gamma$, level k = 1–5) contributes to perceived regularity of a stimulus $R_{ijk}$. For element spacing, the model can be expressed as:

$$R_{ijk} = R^{\alpha}_{i} \tag{Model 2}$$

where $R^{\alpha}_{1} = 0$

One-factor models estimate the contribution of each factor's levels. For example, the above model has three parameters which estimate regularity at the first level ($R^{\alpha}_{1}$), regularity at the second level ($R^{\alpha}_{2}$) and regularity at the third level ($R^{\alpha}_{3}$). For this and the subsequent models, the first level of each factor ($R^{\beta}_{1}$, $R^{\gamma}_{1}$ and $R^{\alpha}_{1}$) is set to 0 in order for the model fits to have a unique solution. Hence, Model 2 has two parameters ($R^{\alpha}_{2-3} = 2$). Similarly, the one-factor models for element size and Jitter levels are:

$$R_{ijk} = R^{\beta}_{j} \tag{Model 3}$$

$$R_{ijk} = R^{\gamma}_{k} \tag{Model 4}$$

where $R^{\beta}_{1}$ and $R^{\gamma}_{1} = 0$, with Model 3 having two parameters ($R^{\beta}_{2-3} = 2$) and Model 4 having four parameters ($R^{\gamma}_{2-5} = 4$), to accommodate the three level values of element size and five jitter levels, respectively.

**Two-additive-factors models.** The two-factors models consider the contribution to regularity of two of the three factors, while ignoring the remaining one. These models assume two factors to contribute to regularity additively and independently from each other. Hence, two-factor-additive models separately estimate the contribution of each factor's levels:

$$R_{ijk} = R^{\alpha}_{i} + R^{\gamma}_{k} \tag{Model 5}$$

$$R_{ijk} = R^{\beta}_{j} + R^{\gamma}_{k} \tag{Model 6}$$

$$R_{ijk} = R^{\alpha}_{i} + R^{\beta}_{j} \tag{Model 7}$$

where $R^{\alpha}_{1}$, $R^{\beta}_{1}$ and $R^{\gamma}_{1} = 0$.

Model 5 and Model 6 have six parameters each ($R^{\alpha}_{2-3}$ or $R^{\beta}_{2-3} + R^{\gamma}_{2-5} = 6$), while Model 7 has four parameters ($R^{\alpha}_{2-3} + R^{\beta}_{2-3} = 4$), again following from the differing numbers of level values for the factors. The three additive models (Models 5–7) will be separately compared

against the three interactive models (Models 8–10) to test for two-way interactions between factors, which are described next.

**Interactive models (Models 8 to 12).   Two-interactive-factors models**

The above models assume the effect of each factor to be additive and independent. However, it is possible that the factors interact with each other in their effect on regularity. For example, the effect of jitter could become stronger when element size is lower. Instead of separately estimating the contribution of each factor's levels, two-interactive-factors models estimate a new parameter for each combination of levels between two factors. For example, Model 8 estimates regularity for each combination of element size and jitter:

$$R_{ijk} = R(\alpha_i, \; \gamma_k) \tag{Model 8}$$

$$R_{ijk} = R(\beta_j, \; \gamma_k) \tag{Model 9}$$

$$R_{ijk} = R(\alpha_i, \; \beta_j) \tag{Model 10}$$

where $R(\alpha_1, \gamma_1)$, $R(\beta_1, \gamma_1)$ and $R(\alpha_1, \beta_1) = 0$.

When both factors are at their first level, we set R to be 0. Models 8 and 9 have a total of fourteen parameters each ($R(\alpha_{1-3}, \gamma_{1-5})$—$R(\alpha_1, \gamma_1) = 14$ for Model 8, and $R(\beta_{1-3}, \gamma_{1-5})$—$R(\beta_1, \gamma_1) = 14$ for Model 9), while Model 10 has eight parameters ($R(\alpha_{1-3}, \beta_{1-3})$—$R(\alpha_1, \beta_1) = 8$), corresponding to the numbers of pairwise combinations of level-values. The three interactive models (Models 8–10) will be compared against the three corresponding additive models (Models 5–7) as described earlier.

**Three-factors two-way interaction model.**   The three-factors two-way interaction model considers all two-way interactions between the three factors:

$$R_{ijk} = R(\beta_j, \; \gamma_k) + R(\alpha_i, \; \gamma_k) + R(\alpha_i, \beta_j) + R(\alpha_i) + R(\beta_j) + R(\gamma_k) \tag{Model 11}$$

where $R = 0$ if any of its factors has a level of 1 (e.g. $R(\alpha_1, \gamma_3) = 0$). This model has a total of 28 parameters ($R(\beta_{2-3}, \gamma_{2-5}) + R(\alpha_{2-3}, \gamma_{2-5}) + R(\alpha_{2-3}, \beta_{2-3}) + R(\alpha_{2-3}) + R(\beta_{2-3}) + R(\gamma_{2-5}) = 28$). However this model omits the possibility of a three-way interaction, so consequently we compare it against the three-factors full model, which is described next, to test for a three-way interaction effect.

**Three-factors full model.**   This is the most complex model. With a total of 44 parameters ($R(\alpha_{1-3}, \beta_{1-3}, \gamma_{1-5})$—$R(\alpha_1, \beta_1, \gamma_1) = 44$), this model attempts to predict the regularity of each possible combination of the three factors, assuming a three-way interaction between the factors:

$$R_{ijk} = R(\alpha_i, \; \beta_j, \; \gamma_k) \tag{Model 12}$$

where $R(\alpha_1, \beta_1, \gamma_1) = 0$.

This model is compared to the three-factors two-way interaction model (Model 11) to test the strength of the three-way interaction.

## Regularity encoding model comparisons

We performed multiple regression analysis using SPSS (IBM Corp., Armonk, NY) to select for the most important SF- or SF × Orientation distribution parameters in Fig 10 that can best predict regularity perception in the experiments (S3 Fig). Stepwise linear regression with forward elimination was used for variable selection. Stepwise linear regression is an iterative procedure where we build a statistical model by adding parameters one by one. On each step, we

add the parameter that most significantly improves the model's prediction of regularity (i.e. the F-test with the smallest p-value). If the smallest p-value is larger than 0.05, no parameter can significantly improve the model's prediction of regularity and the procedure terminates.

## Supporting information

**S1 Fig. Gabor wavelet analysis results for the SF-distribution model.** The same SF analysis as demonstrated in Fig 8 was used here but without mean normalization. See Fig 8 for details.
(TIF)

**S2 Fig. Gabor wavelet analysis results for the SF × orientation distribution model.** The same SF analysis as demonstrated in Fig 9 was performed but without mean normalization. See Fig 9 for details.
(TIF)

**S3 Fig. Perceived regularity estimates from the full model (Model 12) averaged across the five observers.** Color codes for the five jitter levels (yellow to blue). The image-computable models were used to predict the psychophysical results presented here.
(TIF)

**S4 Fig. Gabor wavelet analysis results for Orientation-distribution model.** The same SF analysis as in Fig 8 was performed, except that the energy (ordinate) was normalized to equate total response across the 50 SF levels (Formula 3.2) and was plotted as a function of orientation (abscissa). Smaller element spacing (A), larger element size (B) and lower regularities (C) seem to make the orientation distribution slightly flatter.
(TIF)

**S5 Fig. Gabor wavelet analysis results for the SF-distribution model.** The same SF analysis as in S1 Fig was used, but without initial squaring of input images.
(TIF)

**S6 Fig. Gabor wavelet analysis results for the SF × orientation distribution model on three natural images, with varying degrees of texture regularity.** The same analysis as demonstrated in S2 Fig was performed.
(TIF)

**S7 Fig. Pythagorean distance relative to the grid from the ranges of the five jitter levels (abscissa) to the actual element positions (ordinate).** For each data point (which represents one of the total 45 conditions), the actual element positions are calculated from 184 simple images, with a mean jitter value for each image which was averaged across all the element jitter values within that image. The error bars represent the standard deviation (SD) of the 184 mean jitter values. To check whether element spacing and element size influence the final jitter results, we chose the most different conditions to present here. In A, the smallest spacing (dark line) and largest spacing (light line) at the largest element size level show very little difference in the final jitter positions. Similarly in B, the smallest size (dark line) and largest size (light line) at the smallest element spacing level show very little difference in the final jitter positions. All other conditions were also checked with similar results. This analysis demonstrated that the manipulation of jitter is not significantly confounded with the manipulation of element spacing and size.
(TIF)

**S8 Fig. Parameter estimation of the full model from real data and simulated response.** The input values (abscissa) are the 45 parameters in the full model estimated from the real response

of a naive participant (Observer 4). The output values (ordinate) are the 45 parameters in the full model estimated from simulated response (1035 pairs × 400 times). To simulate the response, the 45 input parameters were used as the means of normal distributions of the 45 experimental conditions. The standard deviation of each distribution was either from the standard deviation (SD) of the final jitter values as described in S7 Fig, which is varied across the 45 conditions and increase with jitter levels (light dots), or is fixed to the mean SD across the 45 conditions (dark dots). Each of the simulated responses of a pair (e.g. condition A vs. condition B) was then decided by randomly choosing a value from the distribution of condition A and condition B separately, and the larger value receives "more regular" response. For all the possible 1035 pairs, the simulation was repeated for 400 times each, and the total 414,000 simulated responses were then used to estimate the 45 parameters in the full model (ordinate). For fixed SDs for all the 45 distributions (dark dots), the parameter estimation of the full model from real response and simulated response are perfectly matched (the dark dots are aligned with the diagonal), reassuring us that our model-fitting procedure is valid. However, when the SDs of distributions are varied (light dots), the parameter estimation from simulated responses showed a slight deviation from the diagonal.
(TIF)

**S1 Table. Deviance values for main effects.** Deviance values (p-values) for the main effect of element spacing (Model 2), size (Model 3) and jitter (Model 4), tested against the Baseline model (Model 1). The p-values were calculated from likelihood-ratio tests (see *Methods* for details).
(DOCX)

**S2 Table. Deviance values for two-way interactions.** Deviance values (p-values) for the interaction effect of element spacing × jitter (Models 5 vs. 8), element size × jitter (Models 6 vs. 9) and element spacing × size (Models 7 vs. 10).
(DOCX)

**S3 Table. Deviance values for three-way interactions.** Deviance values (p-values) for the three-way interaction effect of element spacing × size × jitter (Models 11 vs. 12).
(DOCX)

**S1 Data. Data collected from one participant (out of 5 participants in total, S1-S5), as a.csv file.** Each participant completed 4140 experimental trials, as shown in the 4140 rows: Column 1: response value. 0: participant chose stimulus 1 of a pair as more regular. 1: participant chose stimulus 2 as more regular. Column 2: element spacing level (1–3) for stimulus 1. Column 3: element spacing level (1–3) for stimulus 2. Column 4: element size level (1–3) for stimulus 1. Column 5: element size level (1–3) for stimulus 2. Column 6: element jitter level (1–5) for stimulus 1. Column 7: element jitter level (1–5) for stimulus 2. Note, the trial order was randomized during the experiment.
(CSV)

**S2 Data. Same as S1 Data, but for second participant.**
(CSV)

**S3 Data. Same as S1 Data, but for third participant.**
(CSV)

**S4 Data. Same as S1 Data, but for fourth participant.**
(CSV)

**S5 Data. Same as S1 Data, but for fifth participant.**
(CSV)

## Acknowledgments

We thank our observers for the generous contribution of their time. We are grateful to Prof. Marc Georgeson for stimulating and useful discussions, particularly for the suggestion of an image processing model for regularity perception based on statistics of local orientations.

## Author Contributions

**Conceptualization:** Curtis L. Baker, Jr., Frederick A. A. Kingdom.

**Data curation:** Hua-Chun Sun.

**Formal analysis:** Hua-Chun Sun, David St-Amand.

**Funding acquisition:** Curtis L. Baker, Jr., Frederick A. A. Kingdom.

**Investigation:** Hua-Chun Sun.

**Methodology:** Hua-Chun Sun, David St-Amand, Frederick A. A. Kingdom.

**Project administration:** Hua-Chun Sun.

**Resources:** Curtis L. Baker, Jr., Frederick A. A. Kingdom.

**Software:** Hua-Chun Sun, David St-Amand.

**Supervision:** Curtis L. Baker, Jr., Frederick A. A. Kingdom.

**Validation:** Hua-Chun Sun, David St-Amand.

**Visualization:** Hua-Chun Sun, David St-Amand.

**Writing – original draft:** Hua-Chun Sun, David St-Amand.

**Writing – review & editing:** Hua-Chun Sun, David St-Amand, Curtis L. Baker, Jr., Frederick A. A. Kingdom.

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
