## [Decision Letter · Decision Letter 0]

9 Jun 2021

Dear Dr Baker Jr,

Thank you very much for submitting your manuscript "Visual perception of texture regularity: conjoint measurements and a wavelet response-distribution model" for consideration at PLOS Computational Biology.

As with all papers reviewed by the journal, your manuscript was reviewed by members of the editorial board and by several independent reviewers. In light of the reviews (below this email), we would like to invite the resubmission of a significantly-revised version that takes into account the reviewers' comments.

Dear authors,

Thank you for your submission. I have now read the manuscript myself and secured reviews from two referees, both of whom have provided substantive comments on the text.

Having read the comments made by both reviewers, I agree with them that there are some analytical issues that would need to be addressed in a revision as well as several important theoretical (perhaps it is better to say "big picture") issues that would also be vitally important to take on in some way. Reviewer #2 makes a number of excellent points in their review, so I will not recapitulate everything that they have said in their commentary. Suffice it to say that I think their comments regarding the novelty of some of the analytical tools used here is correct, but I also think that generally speaking your modeling needs to be explained in much clearer terms throughout the paper. The discrepancy between their analysis of Observers 1 and 2 and yours is also something that should be addressed directly. Hopefully this is straightforward given the code they have provided with their review.

It is Reviewer #1's points that I think are more difficult to address as they speak to some core issues in texture perception. Briefly, I agree that the approach used in this study has in large part been superseded by both the use of natural images (and accompanying tools for studying those textures using synthesis algorithms like Portilla-Simoncelli) and the use of more complex artificial images (the reviewer rightly brings up Jonathon Victor's work in this regard). I agree that it is difficult to see this study as especially forward-looking when the key stimulus manipulations involve abstract structured elements. To say something that is perhaps a bit harsh, it isn't clear to me that these results add a great deal to our growing understanding of texture perception because of this limitation. As a result, I'm not entirely confident that even a substantial revision will lead to a publishable manuscript. Still, I don't think it's appropriate to reject the paper outright without giving you the option of attempting to address this important point, so I would like to give you that chance if you choose to do so. To be clear, I think a revision of this paper would have to make a much stronger case than it does currently for what the contribution to new knowledge is, and how the use of these stimuli (1) may lead to generalizable conclusions about a broader class of textures, and (2) provides insights into texture processing that more complex stimuli do not. I think this may be very difficult to do convincingly, but again, I do want to give you the opportunity to try.

Besides these major points, please do also attend to the other suggestions your reviewers have made. Both of their reviews are constructive and thorough, so I hope that you will find their suggestions helpful.

We cannot make any decision about publication until we have seen the revised manuscript and your response to the reviewers' comments. Your revised manuscript is also likely to be sent to reviewers for further evaluation.

Sincerely,

Benjamin Balas

Guest Editor

PLOS Computational Biology

Wolfgang Einhäuser

Deputy Editor

PLOS Computational Biology

Dear authors,

Thank you for your submission. I have now read the manuscript myself and secured reviews from two referees, both of whom have provided substantive comments on the text.

Having read the comments made by both reviewers, I agree with them that there are some analytical issues that would need to be addressed in a revision as well as several important theoretical (perhaps it is better to say "big picture") issues that would also be vitally important to take on in some way. Reviewer #2 makes a number of excellent points in their review, so I will not recapitulate everything that they have said in their commentary. Suffice it to say that I think their comments regarding the novelty of some of the analytical tools used here is correct, but I also think that generally speaking your modeling needs to be explained in much clearer terms throughout the paper. The discrepancy between their analysis of Observers 1 and 2 and yours is also something that should be addressed directly. Hopefully this is straightforward given the code they have provided with their review.

It is Reviewer #1's points that I think are more difficult to address as they speak to some core issues in texture perception. Briefly, I agree that the approach used in this study has in large part been superseded by both the use of natural images (and accompanying tools for studying those textures using synthesis algorithms like Portilla-Simoncelli) and the use of more complex artificial images (the reviewer rightly brings up Jonathon Victor's work in this regard). I agree that it is difficult to see this study as especially forward-looking when the key stimulus manipulations involve abstract structured elements. To say something that is perhaps a bit harsh, it isn't clear to me that these results add a great deal to our growing understanding of texture perception because of this limitation. As a result, I'm not entirely confident that even a substantial revision will lead to a publishable manuscript. Still, I don't think it's appropriate to reject the paper outright without giving you the option of attempting to address this important point, so I would like to give you that chance if you choose to do so. To be clear, I think a revision of this paper would have to make a much stronger case than it does currently for what the contribution to new knowledge is, and how the use of these stimuli (1) may lead to generalizable conclusions about a broader class of textures, and (2) provides insights into texture processing that more complex stimuli do not. I think this may be very difficult to do convincingly, but again, I do want to give you the opportunity to try.

Besides these major points, please do also attend to the other suggestions your reviewers have made. Both of their reviews are constructive and thorough, so I hope that you will find their suggestions helpful.

Reviewer's Responses to Questions

**Comments to the Authors:**

Reviewer #1: In this manuscript, Sun and co-authors investigate the determinants of texture regularity perception. They used an artificial stimulus that can be parametrically controlled and gathered data from five human observers. They found a massive effect of jitter, and comparatively small effects of element spacing and size. They connect these results to an image-processing model and suggest that texture regularity perception is informed by multiple moments of the response distribution of a Gabor filter-bank. I have mixed feelings about this paper. On the one hand, the experiments and modeling are all done well, and the findings are somewhat interesting. On the other hand, the paper approaches the central research question in a surprisingly old-fashioned way and appears divorced from many important research evolutions in the field of texture perception. As I explain below, I think the authors should make a serious effort to better connect their work to recent insights and employ modern tools to achieve this connection.

As argued in the introduction of the paper, the research question is motivated by properties of the natural visual world. However, natural images are statistically very complex and therefore not suited to use as experimental stimuli (they are full of potential confounds). So how does one make these image properties accessible for experimental interrogation? For a long time, the best available option was to create highly artificial stimuli that can easily be controlled, just like the authors did here. The downside of this approach is that these stimuli are so different from natural images that it is not clear whether any knowledge obtained from them transfers to real world perception. About two decades ago, Portilla and Simoncelli opened up a new option by creating a synthesis algorithm that takes natural images as input and produces texture images as output. These synthetic images are much better controlled than natural images, yet much closer to real world vision than a dot lattice. This synthesis tool has become very popular in the vision science community (the original paper has by now gathered over 2000 citations), and has unlocked a lot of new knowledge about texture perception (work by Rosenholtz, Simoncelli, and many others) and its neural and computational basis (e.g., we now know that individual cells in area V2 and V4 are selective for several aspects of visual texture, while neurons in area V1 are not). Jonathan Victor has taken a somewhat different approach, but has also succeeded in directly connecting real-world statistics to texture perception (in collaboration with Hermundstad and many others). The auditory perception community has embraced this synthesis approach as well, resulting in several new insights (McDermott and others). This paper appears divorced from all of this. I find that perplexing.

The issue at stake is profound: would we really care about the documented effects if they only applied to dot lattices? I propose that the authors add a section to their paper in which they confront this question. I suggest they use the Portilla Simoncelli algorithm (or an alternative) to synthesize more naturalistic stimuli that vary along dimensions that are analogous to the ones considered in their experiment. They can then pass these images through their filter bank and perform a similar analysis as they do for the dot lattices to judge whether their identified candidate statistics have any merit beyond this specific stimulus. If the answer comes out positive, it would suggest that these insights have some ecological validity. If the answer comes out negative, then they should think deeply about alternative candidate statistics that would agree with their experimental findings but have larger ecological validity. There are several result sections that can be trimmed and a couple of figures that can be removed if this new section makes the manuscript too long.

Reviewer #2: see attached PDF

**Have the authors made all data and (if applicable) computational code underlying the findings in their manuscript fully available?**

Reviewer #1: Yes

Reviewer #2: Yes

PLOS authors have the option to publish the peer review history of their article (what does this mean?). If published, this will include your full peer review and any attached files.

Reviewer #1: No

Reviewer #2: No
---

## [Decision Letter · Decision Letter 1]

24 Sep 2021

Dear Dr Baker Jr,

We are pleased to inform you that your manuscript 'Visual perception of texture regularity: conjoint measurements and a wavelet response-distribution model' has been provisionally accepted for publication in PLOS Computational Biology.

Best regards,

Benjamin Balas

Guest Editor

PLOS Computational Biology

Wolfgang Einhäuser

Deputy Editor

PLOS Computational Biology

As you will see, your two reviewers have now read the revised manuscript and offered comments on the latest draft. These comments diverge in their evaluation of the paper: Your first reviewer found little change between this draft and the last and offered no new comments as a result. Your second reviewer indicated that they felt the manuscript had adequately addressed the issues they raised previously and offers a minor recommendation for an additional revision/clarification to the text. Having read the revised draft and the comments from both reviewers, my feeling is that there is general agreement that this study remains somewhat limited in scope, but offers some useful insights regarding texture perception within that constrained context. As such, I have decided to accept the manuscript, with the caveat that I would like you to address the remaining comment from your second reviewer (which should be straightforward).

Reviewer's Responses to Questions

**Comments to the Authors:**

Reviewer #1: The paper hasn't changed much, and I don't have much new to say about it.

Reviewer #2: The authors have, in my opinion, done an admirable job of responding to the comments especially with respect to the issue of the use of simplified vs natural stimuli and the complementarity of both types of stimuli in contemporary vision research. All of my comments have been adequately addressed. I only find one small section of confusion in the text on page 30 of the review document, they write,

"For each of the aforementioned comparisons between models, we calculate a deviance value, which represents how many responses are predicted correctly and follows a binomial distribution. This deviance value can be expressed as a likelihood, i.e. a ratio of how many responses are predicted correctly to the total number of responses."

To my understanding, deviance is calculated as minus twice the log likelihood, i.e., the deviance is derived from the likelihood and not vice versa. In addition, the deviance follows or in the current case approximates a chi-squared distribution not a binomial distribution. Here, the likelihood is calculated from the joint probability of a Bernoulli distribution, i.e., a binomial distribution with n = 1, as a function of its parameters, which is what makes it a likelihood rather than a joint probability distribution, but the deviance itself, or the differences of deviances of nested models follow or approximate chi-squared distributions. See https://www.maths.ed.ac.uk/~swood34/core-statistics.pdf p. 83, for example.

**Have the authors made all data and (if applicable) computational code underlying the findings in their manuscript fully available?**

Reviewer #1: Yes

Reviewer #2: Yes

PLOS authors have the option to publish the peer review history of their article (what does this mean?). If published, this will include your full peer review and any attached files.

Reviewer #1: No

Reviewer #2: **Yes: **Kenneth Knoblauch

---

## [Editor Report · Acceptance letter]

11 Oct 2021

PCOMPBIOL-D-21-00121R1 

Visual perception of texture regularity: conjoint measurements and a wavelet response-distribution model

Dear Dr Baker Jr,

I am pleased to inform you that your manuscript has been formally accepted for publication in PLOS Computational Biology. Your manuscript is now with our production department and you will be notified of the publication date in due course.

With kind regards,

Andrea Szabo
